# Task-Aware Structured Memory for Dynamic Multi-modal In-Context Learning

Zhirui Chen[1]  Ziwei Chen[2]  Ling Shao[✉][1]

## Abstract

Multi-modal large language models (MLLMs) depend on in-context learning (ICL) for rapid task adaptation, but their scalability is severely limited by finite context windows and the growing cost of keyvalue (KV) caches in long multi-modal sequences. Existing memory compression approaches typically rely on rigid token removal or sample-dependent importance estimation, which introduces bias, disrupts semantic structureparticularly for visual representationsand yields static memories that cannot adapt to new queries. We introduce TASM (Task-Aware Structured Memory), a training-free framework that addresses these limitations through task-aware, structure-preserving, and dynamically accessible memory construction. TASM employs task-vector guided compression to replace sample-specific signals with a task-level direction that captures shared relevance across demonstrations. To preserve the underlying manifold, it applies semantics-aware token merging via bipartite graph matching, aggregating tokens without destructive pruning. Finally, TASM structures memory into a hierarchy comprising a compact Core Memory and a Latent Bank, facilitating query-adaptive dynamic retrieval. Evaluations confirm TASM maintains high performance under heavy compression, effectively balancing efficiency with adaptability.

## 1. Introduction

In-Context Learning (ICL) has fundamentally changed how Large Language Models (LLMs) are deployed, enabling adaptation to new tasks without parameter updates (Brown

[1]UCAS-Terminus AI Lab, University of Chinese Academy of Sciences, China [2]ByteDance Inc. Correspondence to: Ling Shao <ling.shao@ieee.org>.

*Proceedings of the 43rd International Conference on Machine Learning*, Seoul, South Korea. PMLR 306, 2026. Copyright 2026 by the author(s).

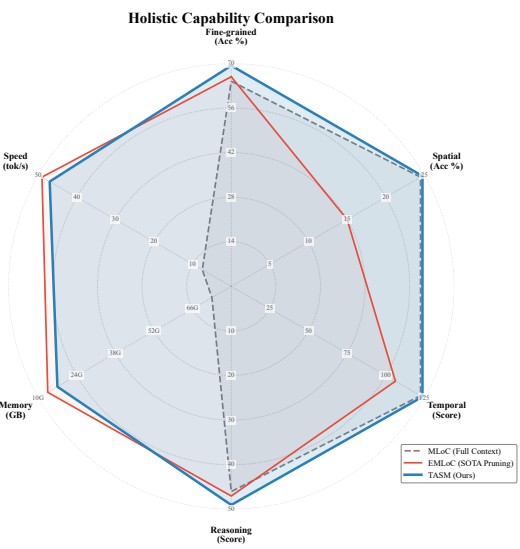

*Figure 1.* Holistic Performance Profile. Comparison of TASM (blue) against Full-Context MLoC (gray) and Pruning-based EM-LoC (red) across six normalized dimensions. While EMLoC sacrifices structural integrity, causing significant degradation in spatial localization and Temporal Reasoning, TASM preserves topological structure, matching full-context capability while maintaining high system efficiency.

et al., 2020). This capability now extends to Multi-modal LLMs (MLLMs), where models like GPT-4V (OpenAI, 2023) and Gemini 3 Pro (Google DeepMind, 2025) demonstrate strong reasoning abilities over interleaved image-text sequences. Recent studies show that performance scales log-linearly with the number of demonstrations (Jiang et al., 2024; Agarwal et al., 2024). This "Many-Shot" regime offers a compelling alternative to fine-tuning, particularly for tasks with scarce data (Zhang et al., 2024a).

However, Many-Shot ICL incurs high computational costs (Wan et al., 2025). Processing thousands of images generates massive Key-Value (KV) caches that overwhelm GPU memory and increase latency. While cache compression strategies effectively support text-only models (Zhang et al., 2023; Xiao et al., 2024; Li et al., 2024b), these methods often fail in the multimodal domain. As illustrated in Figure 1, while existing methods like EMLoC optimize for efficiency, they suffer significant performance capability gaps in spatial

and temporal tasks.

Current multimodal compression frameworks typically rely on text-based heuristics, such as pruning based on answer-aware attention scores (Wang et al., 2024a). We argue that these approaches suffer from three structural limitations. First, *sample-specific bias*: using attention scores from specific demonstration answers overfits to those examples, often removing context that is irrelevant to the demonstration but crucial for the test query (Lin et al., 2025). Second, *topological destruction*: visual tokens possess inherent spatial and semantic structure (Dosovitskiy et al., 2020). Hard pruning strategies, such as Top-K selection, disrupt the 2D information manifold and destroy spatial relationships needed for tasks like localization (Shahgir et al., 2024). Third, *static memory rigidity*: once a cache is compressed, it becomes static. The model cannot recover discarded information when it encounters a complex query requiring fine-grained details (Sharma & Sharma, 2024).

To address these challenges, we introduce TASM (Task-Aware Structured Memory). TASM decouples memory compression from specific demonstration samples. Instead of treating the KV cache as a static buffer, we view it as a structured representation of the task. Our approach guides compression using the transformation direction of the task itself rather than the content of individual samples (Huang et al., 2024). TASM introduces three key innovations:

- **Task-Vector Guided Compression:** Instead of relying on answer-specific attention, we extract a global "task vector" (Huang et al., 2024) that represents the transformation from questions to answers. We compute importance scores by projecting embeddings onto this direction, ensuring the memory captures general reasoning patterns rather than sample biases.

- **Semantics-Aware Token Merging:** To preserve spatial structure, we replace hard pruning with a soft merging mechanism. Using bipartite graph matching, we aggregate low-importance tokens into semantic clusters. This retains the spatial layout and information density of the original images.

- **Query-Adaptive Dynamic Activation:** We propose a hierarchical architecture with a compressed *Core Memory* and a larger, CPU-offloaded *Latent Bank*. During inference, a retrieval module dynamically fetches relevant tokens from the Latent Bank based on the query, allowing the memory to adapt to the complexity of the test instance.

Evaluations on benchmarks such as IllusionVQA (Shahgir et al., 2024), MME-RealWorld (Zhang et al., 2024b), and ImageNet-100 (Deng et al., 2009) demonstrate that TASM significantly outperforms existing baselines. We achieve performance comparable to full-context Many-Shot ICL while reducing memory usage by up to 80%, effectively enabling robust long-context adaptation on consumer-grade hardware.

## 2. Related Work

### 2.1. Multimodal Large Language Models (MLLMs)

The evolution of Multimodal Large Language Models has rapidly shifted from specialized few-shot learners to general-purpose foundation models. Proprietary systems such as GPT-4 (Achiam et al., 2023) and Gemini (Google Deep-Mind, 2025) currently define the state-of-the-art, demonstrating exceptional reasoning across diverse modalities. Concurrently, the open-source community has democratized access through models like LLaVA (Liu et al., 2024), Qwen-VL (Bai et al., 2023), and MiniCPM (Hu et al., 2024), which bridge powerful LLM backbones with visual encoders to handle complex visual instructions. Recent iterations, such as LLaVA-NeXT (Li et al., 2024a), further enhance these capabilities through improved dynamic resolution and reasoning logic. However, despite their impressive performance, these architectures face a fundamental bottleneck: the computational cost of the attention mechanism grows linearly or quadratically with input length. This scaling limitation severely constrains their ability to process long-context histories, a critical requirement for dynamic, real-world applications (Chen et al., 2025).

### 2.2. In-Context Learning and Many-Shot Scaling

In-Context Learning (ICL) has emerged as a paradigm to adapt models to new tasks without parameter updates. While early research focused on few-shot scenarios, recent empirical studies, such as MLOC (Jiang et al., 2024) and the work by (Agarwal et al., 2024), validate the "Many-Shot" hypothesis: scaling the number of demonstration examples into the hundreds or thousands yields significant performance gains, particularly for complex reasoning tasks. To manage the complexity of these tasks, researchers have begun exploring "Task Vectors" (Huang et al., 2024), which aggregate task-specific information into compact representations. It is crucial to distinguish our approach from prior work; while (Huang et al., 2024) utilizes task vectors primarily for steering model weights during inference, TASM uniquely applies this concept to *memory management*. We leverage task vectors to estimate the importance of Key-Value (KV) pairs, enabling precise compression of the context cache rather than modifying model parameters.

### 2.3. Efficient Inference and KV Cache Compression

To address the memory overhead of long-context generation, various efficient inference techniques have been pro-

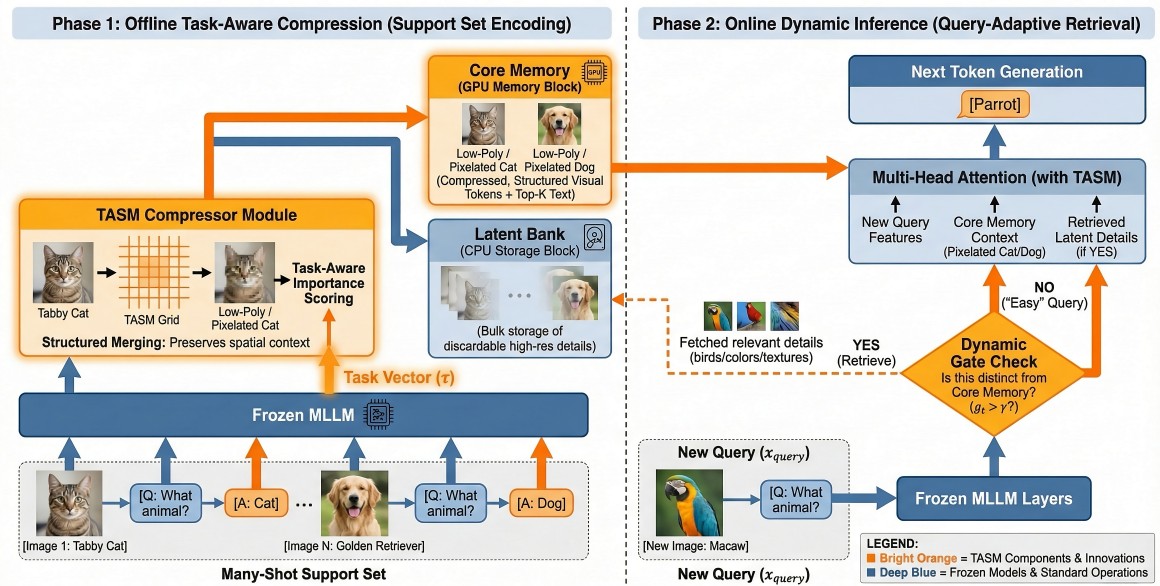

*Figure 2.* TASM: Task-Aware Structured Memory Framework (Training-Free). This figure illustrates the two-phase methodological framework. Left Panel (Phase 1): Offline compression encodes the support set. A frozen MLLM extracts a global task vector ($\tau$). The TASM compressor module uses $\tau$ for importance scoring and performs semantics-aware token merging (in latent KV-space, not pixel space) to create compact tokens for the GPU-resident core memory, while discarding high-res tokens to the CPU-resident latent bank. Right Panel (Phase 2): Online inference uses a dynamic gate based on JS divergence to determine if a new query requires retrieving additional context from the latent bank before final generation via multi-head attention. The entire pipeline requires no parameter updates.

posed. In the text-only domain, token pruning methods like H2O (Zhang et al., 2023), SnapKV (Li et al., 2024b), and PyramidKV (Cai et al., 2024) reduce memory footprint by discarding less attentive tokens ("hard pruning"). However, applying these text-centric heuristics directly to multimodal data is problematic, as hard pruning often disrupts the spatial topology and structural coherence required to interpret visual tokens effectively.

In the multimodal landscape, EMLoC (Ma et al., 2025) represents a primary baseline, attempting to extend long-context capabilities via efficient retrieval. However, EM-LoC relies on "Answer-Aware Attention," which introduces potential bias by conditioning retrieval on proxy answers, and employs hard pruning that risks severing critical visual dependencies. TASM addresses these limitations by replacing hard pruning with soft token merging, guided by task-agnostic vectors. This approach in Figure 2 preserves the structural integrity of visual memory while achieving high compression ratios, ensuring that dynamic, many-shot ICL remains both efficient and robust.

## 3. Methodology

We introduce **Task-Aware Structured Memory (TASM)**, a framework designed to resolve the fundamental memory bottleneck in Multi-modal Large Language Models (MLLMs). Standard KV cache compression techniques typically rely on attention accumulation scores, implicitly

assuming that past attention weights are sufficient predictors of future relevance. We challenge this assumption, arguing that importance is fundamentally *task-dependent* and that hard pruning destroys the semantic topology of multi-modal features.

TASM reformulates memory compression through three coupled theoretical innovations:

(1) **Task-Vector Guided Importance**, which projects token representations onto a latent task manifold to separate signal from noise; (2) **Semantics-Aware Token Merging**, which treats compression as a spatially-constrained graph matching problem to preserve visual topology; (3) **Information-Geometric Dynamic Retrieval**, which enables adaptive memory expansion driven by distributional shifts in the attention mechanism. Pseudo code of TASM is presented in Appendix A.

### 3.1. Preliminaries and Problem Formulation

Consider an autoregressive MLLM processing a sequence $\mathbf{X} = \{x_1, \ldots, x_t\}$ containing both discrete text tokens and continuous visual patches. At layer $l \in [1, L]$, the model maps the input to hidden states $\mathbf{H}_l \in \mathbb{R}^{t \times d_{\text{model}}}$. The multi-head attention mechanism computes queries, keys, and values for head $h$ via linear projections:

$$\mathbf{Q}_{t,l}^h = \mathbf{x}_{t,l} \mathbf{W}_Q^h, \quad \mathbf{K}_{t,l}^h = \mathbf{H}_{t,l} \mathbf{W}_K^h, \quad \mathbf{V}_{t,l}^h = \mathbf{H}_{t,l} \mathbf{W}_V^h \tag{1}$$

where $\mathbf{W}_Q, \mathbf{W}_K, \mathbf{W}_V \in \mathbb{R}^{d_{\text{model}} \times d_k}$. To avoid quadratic recomputation, the model maintains a Key-Value (KV) cache $\mathcal{C}_t$, formally defined as the union of states across all layers:

$$\mathcal{C}_t = \bigcup_{l=1}^{L} \{(\mathbf{K}_{\leq t,l}, \mathbf{V}_{\leq t,l})\} \tag{2}$$

The memory complexity of $\mathcal{C}_t$ scales linearly with time step $t$, creating a bottleneck for long-context reasoning. The space complexity is given by:

$$\text{Space}(\mathcal{C}_t) \in \mathcal{O}(T \cdot L \cdot H \cdot d_k) \tag{3}$$

where $T$ is the context length. We seek a compression operator $\Phi_t : \mathcal{C}_t \to \hat{\mathcal{C}}_t$ that enforces a strict budget constraint $|\hat{\mathcal{C}}_t| \leq B \ll T$ while minimizing the information loss. This is formalized as minimizing the Kullback-Leibler divergence between the predictive distributions of the full and compressed contexts:

$$\min_{\Phi} \mathbb{E}_{x_{t+1} \sim \mathcal{D}} \left[ D_{\text{KL}}\Big( P_\theta(x_{t+1} \mid \mathcal{C}_t) \,\big\|\, P_\theta(x_{t+1} \mid \Phi_t(\mathcal{C}_t)) \Big) \right] \tag{4}$$

### 3.2. Task-Vector Guided Importance Estimation

A critical limitation of attention-based importance metrics (e.g., H2O, EMLoC) is their susceptibility to "attention sinks"—tokens with high attention scores but low semantic value (e.g., punctuation). We propose that true semantic importance is defined not by generic attention, but by *alignment with the task transformation direction*.

**Task Vector Extraction via Few-Shot Geometry.** We leverage the in-context learning setup where the context contains few-shot demonstrations. Let $\mathcal{S}_{\text{few}} = \{(\mathbf{Q}^{(n)}, \mathbf{A}^{(n)})\}_{n=1}^{N}$ be the set of Question-Answer pairs. We hypothesize that the transformation from question to answer represents a distinct, coherent vector field in the activation space. We compute the semantic centroids for the Question and Answer representations at layer $l$ as:

$$\boldsymbol{\mu}_{\mathbf{Q},l} = \frac{1}{Z_Q} \sum_{n=1}^{N} \sum_{j \in \text{span}(Q)} \mathbf{h}_{j,l}^{(Q,n)} \tag{5}$$

$$\boldsymbol{\mu}_{\mathbf{A},l} = \frac{1}{Z_A} \sum_{n=1}^{N} \sum_{j \in \text{span}(A)} \mathbf{h}_{j,l}^{(A,n)} \tag{6}$$

The *Task Vector* $\boldsymbol{\tau}_l \in \mathbb{R}^{d_k}$ is derived as the normalized difference vector, encoding the direction of reasoning (e.g., "visual extraction" or "logic inference"):

$$\boldsymbol{\tau}_l = \frac{\boldsymbol{\mu}_{\mathbf{A},l} - \boldsymbol{\mu}_{\mathbf{Q},l}}{\|\boldsymbol{\mu}_{\mathbf{A},l} - \boldsymbol{\mu}_{\mathbf{Q},l}\|_2} \tag{7}$$

**Orthogonal Projection Scoring.** To quantify the relevance of a cached token, we project its key representation onto the task vector. This effectively filters out noise that is orthogonal to the task direction. The task-based score $s_{i,l}^{\text{task}}$ is defined as:

$$s_{i,l}^{\text{task}} = \text{ReLU}\left( \frac{\mathbf{k}_{i,l}^\top \boldsymbol{\tau}_l}{\|\mathbf{k}_{i,l}\|} \right) + \gamma \frac{\|\mathbf{v}_{i,l}\|_2}{\max_j \|\mathbf{v}_{j,l}\|_2} \tag{8}$$

Here, the ReLU ensures we only prioritize tokens positively aligned with the task, and the value-norm term captures the magnitude of information content.

**Layer-Adaptive Gating Mechanism.** Transformer layers exhibit functional specialization: shallow layers process local, high-frequency features (syntax, edges), while deep layers process global semantics. Relying solely on the task vector in shallow layers ignores necessary local context. We introduce a layer-adaptive gating function $\lambda(l)$ to interpolate between local attention scores ($s^{\text{attn}}$) and global task scores ($s^{\text{task}}$):

$$\mathcal{S}_{i,l} = \lambda(l) \cdot s_{i,l}^{\text{task}} + (1 - \lambda(l)) \cdot s_{i,l}^{\text{attn}} \tag{9}$$

We model $\lambda(l)$ using a shifted sigmoid function to create a smooth transition from local to global focus across the network depth:

$$\lambda(l) = \alpha + \beta \cdot \sigma\left( \kappa \cdot \left( \frac{l}{L} - 0.5 \right) \right) \tag{10}$$

Empirically setting $\alpha = 0.1, \beta = 0.8, \kappa = 10$ results in $\lambda(l) \approx 0.1$ for shallow layers (attention dominant) and $\lambda(l) \approx 0.9$ for deep layers (task vector dominant), establishing a hierarchical importance metric.

### 3.3. Semantics-Aware Token Merging

Traditional methods utilize *Hard Top-K Pruning*, defined as $\hat{\mathcal{C}} = \{c_i \mid \text{rank}(S_i) \leq K\}$. While efficient, this approach is destructive for visual representations where information is distributed across spatially adjacent patches. Removing "redundant" background patches breaks the 2D manifold structure required by convolution-like attention heads. We propose *Semantics-Aware Token Merging*, treating compression as a graph matching problem.

**Graph Formulation.** We partition the token set $\mathcal{V}_l$ into *Sink* nodes $\mathcal{U}_{\text{sink}}$ (high-scoring tokens to be preserved) and *Source* nodes $\mathcal{U}_{\text{src}}$ (low-scoring tokens to be merged). We construct a bipartite graph $\mathcal{G} = (\mathcal{U}_{\text{src}}, \mathcal{U}_{\text{sink}}, \mathcal{E})$ and seek an optimal assignment matrix $\mathbf{M} \in \{0,1\}^{|\mathcal{U}_{\text{src}}| \times |\mathcal{U}_{\text{sink}}|}$. The optimization objective is to maximize the retained semantic information under the constraint that each source merges into at most one sink:

$$\max_{\mathbf{M}} \sum_{i \in \mathcal{U}_{\text{src}}} \sum_{j \in \mathcal{U}_{\text{sink}}} M_{ij} \cdot w_{ij}, \quad \text{s.t.} \sum_j M_{ij} \leq 1 \tag{11}$$

**Spatially-Constrained Compatibility.** The edge weight $w_{ij}$ measures the compatibility between tokens. For text, this is simply cosine similarity. However, for visual tokens with spatio-temporal coordinates $\mathbf{p} = (t, h, w)$, we must impose a locality constraint to prevent physically distant patches from merging. We define a spatial regularizer $\Psi$:

$$\Psi(i,j) = \begin{cases} 0 & \text{if } \|\mathbf{p}_i - \mathbf{p}_j\|_1 \leq \Delta_{\text{win}} \\ -\infty & \text{otherwise} \end{cases} \quad (12)$$

where $\Delta_{\text{win}}$ is the receptive field size. The unified compatibility metric is:

$$w_{ij} = \cos(\mathbf{k}_i, \mathbf{k}_j) + \Psi(i,j) \cdot \mathbb{I}(i,j \in \mathcal{V}_{\text{visual}}) \quad (13)$$

This forces visual tokens to merge only with their spatial neighbors, preserving the local feature geometry essential for vision tasks.

**Manifold-Preserving Aggregation.** To approximate the optimal transport of information from source to sink, we employ a weighted aggregation update. The compressed state of a sink token $j$ becomes the weighted centroid of its cluster:

$$\hat{\mathbf{k}}_j = \frac{1}{Z_j} \left( \mathbf{k}_j + \sum_{i \in \mathcal{U}_{\text{src}}} M_{ij} \cdot e^{w_{ij}} \cdot \mathbf{k}_i \right) \quad (14)$$

This operation effectively acts as a dynamic pooling layer, condensing information rather than discarding it.

### 3.4. Information-Geometric Dynamic Retrieval

To handle infinite contexts within a fixed memory budget $B$, TASM implements a hierarchical memory system comprising a high-bandwidth **Core Memory** ($\mathcal{M}_{\text{core}}$) and a high-capacity **Latent Bank** ($\mathcal{M}_{\text{latent}}$).

**The Retrieval Trigger: JS Divergence.** Static retrieval policies (e.g., periodic retrieval) are computationally wasteful. We propose a trigger based on the *stability of the attention distribution*. Let $P_{\text{ref}}$ be the attention distribution over $\mathcal{M}_{\text{core}}$ at step $t-1$, and $P_t$ be the distribution at step $t$. We measure the distributional shift using the symmetric Jensen-Shannon (JS) Divergence:

$$D_{\text{JS}}(P_t \| P_{\text{ref}}) = \frac{1}{2} D_{\text{KL}}(P_t \| \mathcal{D}) + \frac{1}{2} D_{\text{KL}}(P_{\text{ref}} \| \mathcal{D}) \quad (15)$$

where $\mathcal{D} = \frac{1}{2}(P_t + P_{\text{ref}})$ is the mixture distribution. A high divergence implies that the current query is attending to a novel subspace, signaling a need to retrieve from the Latent Bank.

**Sparse Retrieval Mechanism.** We define a binary retrieval gate $g_t = \mathbb{I}(D_{\text{JS}} > \epsilon)$. When triggered ($g_t = 1$), we execute a top-$k$ retrieval against the Latent Bank to fetch relevant historical tokens:

$$\mathcal{K}_{\text{retrieved}} = \text{Top-}k \left( \frac{\mathbf{q}_t \mathbf{K}_{\text{latent}}^\top}{\sqrt{d_k}} \right) \quad (16)$$

The effective cache for the current step is dynamically expanded:

$$\mathcal{C}_t = \mathcal{M}_{\text{core}} \cup \mathcal{K}_{\text{retrieved}} \quad (17)$$

This mechanism allows TASM to adaptively access long-term history only when semantically necessary, maintaining $\mathcal{O}(1)$ average complexity for the majority of generation steps.

### 3.5. Complexity Analysis

TASM achieves quasi-linear scaling with sequence length. Task Vector extraction is performed once per task ($\mathcal{O}(1)$). The greedy implementation of bipartite matching requires $\mathcal{O}(T \log T)$ for sorting and $\mathcal{O}(T)$ for assignment. During inference, given core memory size $N_{\text{core}}$ and retrieved size $N_{\text{ret}}$, the attention complexity reduces from $\mathcal{O}(T^2)$ to:

$$\mathcal{O}(T \cdot (N_{\text{core}} + N_{\text{ret}})) \quad (18)$$

Since $(N_{\text{core}} + N_{\text{ret}}) \ll T$ is bounded by a constant budget, TASM enables efficient inference on commodity hardware with limited VRAM.

## 4. Experiments

### 4.1. Experimental Setting

**Evaluation Datasets.** To strictly evaluate the versatility and robustness of TASM, we conduct experiments across a comprehensive suite of nine vision-language benchmarks covering diverse modalities. For fine-grained visual recognition, we use ImageNet-100 (Tian et al., 2020) and IllusionVQA (Shahgir et al., 2024). To assess spatial reasoning and topology preservation, we utilize ScreenSpot (Cheng et al., 2024) for GUI element grounding and RefCOCOg (Kazemzadeh et al., 2014) for visual grounding. For complex reasoning and knowledge retrieval, we employ OK-VQA (Marino et al., 2019) and MME-RealWorld (MME-RW) (Zhang et al., 2024b). To evaluate long-context and temporal capabilities, we include YouCook2 (Zhou et al., 2018) for video captioning and the more challenging VideoMME (without subtitles) (Fu et al., 2025) for long-video understanding. Furthermore, we introduce the Visual Needle-in-a-Haystack (V-NIAH) (Wu et al., 2025) task to rigorously stress-test the retrieval of fine-grained visual details from extensive context.

**Implementation Details.** We primarily select Qwen2-VL-7B-Instruct (Wang et al., 2024b) as our base MLLM due to

*Table 1.* Main results of TASM across diverse multi-modal benchmarks. The values in green indicate the average context length. † denotes 50 examples; ‡ denotes 200 examples.

| Method | Example Number | ImageNet100 | ScreenSpot | MME-RW | IllusionVQA | OK-VQA | YouCook2 |
|---|---|---|---|---|---|---|---|
| Llava1.5 (7B) | | 12.3 | 9.7 | 28.2 | 24.1 | 53.6 | - |
| InternVL2 (8B) | | 12.5 | 2.7 | 33.9 | 28.0 | 47.1 | 88.0 |
| Llama3.2-V (11B) | 0 | 47.6 | 8.1 | 14.6 | 33.0 | - | - |
| MiniCPM-V2.6 (8B) | | 31.0 | 0.3 | 37.2 | 34.6 | 48.3 | 3.3 |
| Qwen2-VL (7B) | | 28.0 | 14.2 | 36.6 | 35.3 | 52.1 | 25.4 |
| MLoC (Qwen2-VL) | 5 | 43.2 † | 14.7 | 39.4 | 38.8 | 58.4 | 86.9 |
| | | 4109 | 1996 | 1924 | 1826 | 1401 | 5907 |
| | 20 | 62.6 ‡ | 18.2 | 41.1 | 40.9 | 58.6 | 108.8 |
| | | 16264 | 7905 | 7393 | 7271 | 5730 | 23464 |
| EMLoC (Qwen2-VL) | 20 | 63.6 ‡ | 18.3 | 42.2 | 40.9 | 58.7 | 102.0 |
| | | 3643 | 1415 | 1510 | 1878 | 934 | 6218 |
| **TASM (Ours)** | 20 | **65.0 ‡** | **19.5** | **43.5** | **42.0** | **60.1** | **109.5** |
| | | **3485** | **1268** | **1437** | **1757** | **843** | **6060** |

its native support for dynamic resolution and long contexts. To demonstrate architectural generalization, we also evaluate on LLaVA-NeXT-Video (7B) and InternVL3 (8B) in Appendix B.4. All experiments are conducted on 4 NVIDIA A800 GPUs (80GB VRAM). Hyperparameters are derived from our empirical ablations. (1) We employ "combined" task-vector extraction with an interpolation weight $\alpha = 0.3$. (2) For semantics-aware merging, we enforce a local spatial window of $3 \times 3$ ($\Delta_{win} = 3$) with a similarity threshold of 0.5. (3) The hierarchical memory allocates 20% to the active Core Memory and 40% to the Latent Bank. (4) Dynamic retrieval is triggered when the JS divergence exceeds $\delta = 0.002$, fetching the top-96 relevant tokens. We compare TASM against Full-Context inference and state-of-the-art compression methods including EMLoC (Ma et al., 2025), SnapKV (Li et al., 2024b), FastV (Chen et al., 2024), and SparseVLM (Zhang et al., 2025).

## 4.2. Experimental Results

**Performance of TASM on multiple tasks.** As presented in Table 1, our proposed Task-Aware Structured Memory (TASM) framework consistently outperforms both the vanilla multi-modal long-context learning (MLoC) and the state-of-the-art compression method EMLoC across all evaluated benchmarks. While MLoC significantly enhances Qwen2-VL's baseline capabilities by leveraging extensive demonstrations, it incurs prohibitive memory costs.

TASM effectively addresses this bottleneck while achieving superior accuracy. When utilizing 200 examples, TASM dramatically reduces the average context length on ImageNet100 from *16264* (MLoC) to *3485*, a remarkable 78.6% reduction. Crucially, unlike EMLoC, which suffers performance degradation on complex temporal tasks like YouCook2 (dropping from 108.8 to 102.0), TASM not only recovers this loss but surpasses the full-context baseline, achieving a score of 109.5. Notably, TASM achieves

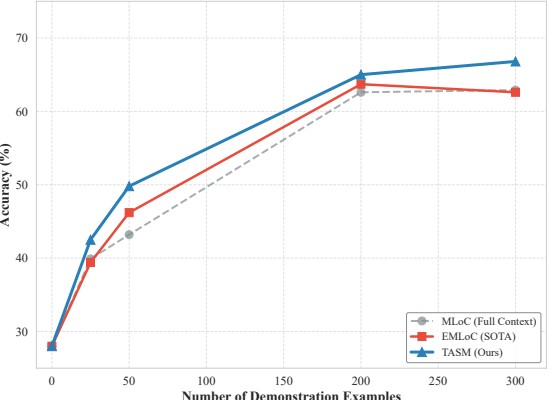

*Figure 3.* Many-Shot Scaling Law on ImageNet-100.

this superior performance with a lower memory footprint than EMLoC (*6060* vs. *6218* tokens), validating that our dynamic retrieval mechanism is more efficient at recalling fine-grained temporal details than static pruning.

Furthermore, in spatially sensitive tasks such as ScreenSpot, TASM achieves a clear margin over EMLoC (19.5 vs. 18.3). This improvement confirms our hypothesis that EMLoC's hard pruning disrupts the visual information manifold, whereas TASM's semantics-aware token merging preserves the essential topological structure of visual features. By filtering out task-irrelevant noise via the task vector while retaining semantic density, TASM yields a more precise memory representation, enabling robust performance even under extreme compression ratios. More results on Qwen2.5-VL-7B-Instruct are presented in Appendix B.2.

**Results with varying numbers of examples.** Figure 3 illustrates the performance trajectory of MLoC, EMLoC, and TASM as the number of demonstration examples scales from 0 to 300 on ImageNet-100. While all methods exhibit performance gains in the early stages, a distinct divergence

emerges at higher shot counts. Notably, the state-of-the-art compression method EMLoC exhibits performance saturation and slight degradation when scaling from 200 to 300 examples (dropping from 63.7% to 62.6%). This indicates that rigid pruning heuristics may inadvertently discard critical semantic details in extremely long contexts. In contrast, TASM maintains a robust log-linear scaling behavior, consistently outperforming both the full-context MLoC and EMLoC. At 300 examples, TASM achieves a peak accuracy of 66.8%, surpassing EMLoC by a significant margin. This confirms that TASM's structure-preserving merging and dynamic retrieval enable the model to effectively assimilate information from massive demonstrations without suffering from the "context saturation" observed in prior methods.

Table 2. Visual Needle-in-a-Haystack (V-NIAH) Performance.

| Method | # Input Images (Haystack Size) | | | | Avg. |
|---|---|---|---|---|---|
| | 10 | 50 | 100 | 200 | |
| *Full Context* | 96.5 | 94.2 | *OOM* | *OOM* | - |
| SparseVLM | 85.2 | 42.1 | 15.6 | 8.4 | 37.8 |
| FastV | 82.4 | 38.5 | 12.3 | 7.1 | 35.1 |
| EMLoC | 91.0 | 68.3 | 35.2 | 18.9 | 53.4 |
| **TASM (Ours)** | 95.8 | 80.4 | 49.1 | 45.5 | 67.7 |

**Visual Needle-in-a-Haystack.** Table 2 presents the results on the V-NIAH task, which demands the retrieval of a specific visual detail (the "needle") from a long sequence of distractors. Traditional pruning methods like FastV and SparseVLM exhibit a catastrophic performance drop as the haystack size increases (e.g., FastV collapses to 7.1% at 200 images). This confirms that static pruning strategies, which discard tokens based on instantaneous saliency, permanently lose information that appears irrelevant during pre-filling but becomes crucial for the final query. Similarly, EMLoC struggles in deep contexts (18.9% at 200 images) because its answer-aware heuristic overfits to the demonstration patterns, neglecting unique details in the test input. In contrast, TASM demonstrates superior resilience against forgetting. At the most challenging setting of 200 images, TASM maintains an accuracy of 45.5%, more than doubling the performance of EMLoC (18.9%) and outperforming FastV by over 6×. This substantial margin validates the efficacy of our dynamic retrieval mechanism: instead of permanently deleting the "needle," TASM compresses it into the latent bank and retrieves these dormant features via the JS divergence trigger, effectively mitigating the "memory rigidity" problem inherent in prior works. Detailed results on VideoMME are provided in Appendix B.1.

### 4.3. Ablation Studies

**Systemic Factorial Ablation.** To quantify the marginal contributions of each proposed innovation, we conducted a progressive ablation across three distinct capability dimen-

sions, as shown in Table 3. Task Vector Scoring effectively filters noise, boosting the baseline significantly (e.g., +9.9% on ImageNet-100). Semantics-Aware Token Merging prevents topological destruction; enforcing spatial constraints massively improves ScreenSpot (+6.7%) by preserving the 2D manifold. Finally, Dynamic Retrieval resolves static memory rigidity, delivering a +11.0% gain on YouCook2, confirming its necessity for recalling dormant temporal details.

Table 3. Systemic Factorial Ablation of TASM Components. All settings use a 20% memory budget.

| Baseline (Hard Pruning) | + Task Vector Scoring | + Token Merging | + Dynamic Retrieval | IN-100 (General) | ScreenSpot (Spatial) | YouCook2 (Temporal) |
|---|---|---|---|---|---|---|
| ✓ | | | | 45.3 | 10.5 | 85.2 |
| ✓ | ✓ | | | 55.2 | 12.1 | 92.4 |
| ✓ | ✓ | ✓ | | 61.8 | 18.8 | 98.5 |
| ✓ | ✓ | ✓ | ✓ | **65.0** | **19.5** | **109.5** |

**Impact of Layer-Adaptive Gating.** In MLLMs, shallow layers primarily extract high-frequency visual geometry, while deep layers handle task-specific reasoning. We compared our adaptive schedule $\lambda(l)$ against a Uniform Gating baseline (fixed $\lambda = 0.5$). As shown in Table 4, forcing task-vector dominance in shallow layers (Uniform Gating) destroys spatial encoding, causing a 3.3% drop on ScreenSpot. Our layer-adaptive schedule resolves this by preserving local visual attention early on and guiding task reasoning in deeper layers. Furthermore, the exact values of the hyperparameters $(\alpha, \beta, \kappa)$ in Eq. 10 are highly robust as long as this general trend is maintained.

Table 4. Ablation on Layer-Adaptive Gating Schedule.

| Gating Strategy | $\lambda$ in Shallow Layers | $\lambda$ in Deep Layers | ScreenSpot | MME-RW |
|---|---|---|---|---|
| Uniform Gating | 0.5 | 0.5 | 16.2 | 41.8 |
| **Adaptive (Ours)** | $\approx$**0.1** | $\approx$**0.9** | **19.5** | **43.5** |

**Impact of Semantics-Aware Token Merging.** Table 5 compares the structural integrity of different mechanisms. On semantic classification (ImageNet-100), TASM maintains superior accuracy (60.8%) compared to EMLoC (57.7%) at a 10% retention ratio, demonstrating the benefit of feature aggregation. The contrast is more pronounced in spatially sensitive tasks like ScreenSpot and RefCOCOg, where hard pruning incurs catastrophic information loss. Specifically, EMLoC drops to 12.5 on ScreenSpot, indicating that pruning disrupts the visual manifold by discarding "background" patches essential for coordinates. In contrast, TASM uses bipartite matching to merge tokens, significantly mitigating degradation. It achieves 15.9 on ScreenSpot and 67.4 on RefCOCOg (outperforming EMLoC by +3.4% and +4.6%), proving that soft merging preserves spatial cues lost by static pruning.

**Impact of Task-Vector Guided Scoring.** Table 6 compares different importance scoring mechanisms. Standard attention (e.g., H2O) biases heavily towards "attention sinks,"

*Table 5.* Ablation of Compression Mechanisms: Merging vs. Pruning.

| Method | Ratio | Classification | Spatial Localization | |
|---|---|---|---|---|
| | | ImageNet-100 | ScreenSpot | RefCOCOg |
| *Full Context* | 100% | 62.6 | 19.8 | 78.5 |
| SnapKV | | 47.6 | 8.5 | 42.1 |
| FastV | 20% | 52.3 | 12.4 | 55.6 |
| EMLoC | | 63.6 | 18.3 | 75.2 |
| **TASM (Ours)** | | **65.0** | **19.5** | **77.1** |
| SnapKV | | 42.1 | 4.2 | 28.3 |
| FastV | 10% | 45.8 | 8.1 | 41.5 |
| EMLoC | | 57.7 | 12.5 | 62.8 |
| **TASM (Ours)** | | **60.8** | **15.9** | **67.4** |

*Table 6.* Ablation on Importance Scoring.

| Method | Retain Quality | | Performance | | | Avg. |
|---|---|---|---|---|---|---|
| | Vis.%↑ | Sink%↓ | IN-100 | MME | OK-VQA | |
| Random | 78.5 | 2.1 | 15.2 | 10.5 | 12.8 | 12.8 |
| Attn. (H2O) | 45.2 | 38.6 | 47.6 | 39.6 | 48.2 | 45.1 |
| Ans-Aware (EMLoC) | 68.4 | 15.3 | 63.6 | 42.2 | 58.7 | 54.8 |
| **Task-Vec (Ours)** | **79.2** | **5.4** | **65.0** | **43.5** | **60.1** | **56.2** |

wasting 38.6% of memory on non-semantic tokens. Even EMLoC's answer-aware approach retains significant syntactic noise (15.3%), limiting the capacity for visual features. In contrast, our Task-Vector projection effectively disentangles signal from noise, improving the *Visual Ratio* to 79.2% while reducing noise to 5.4%a 3× reduction compared to EMLoC. This structural purity translates into consistent gains, boosting accuracy by 1.3%–1.4% across benchmarks, confirming that task-aligned scoring provides a significantly higher-quality memory signal. See Appendix B.3 for an analysis of sample efficiency and construction cost.

**Efficacy of Dynamic Retrieval.** Table 7 evaluates the impact of different retrieval policies on long-context adaptation. The static core baseline, which relies solely on compressed memory without retrieval, suffers significant performance drops (e.g., 92.4 on YouCook2), highlighting the necessity of accessing historical details stored in the Latent Bank. While periodic-dense retrieval restores performance (108.9), it incurs a high computational overhead with a 10% trigger rate and 45ms latency, as it blindly retrieves information even when the local context is sufficient. Conversely, periodic-sparse improves efficiency but fails to capture transient dependencies, leading to suboptimal accuracy. TASM demonstrates superior intelligence by leveraging JS divergence as a surprise signal. It achieves a state-of-the-art score of 109.5 on YouCook2 with a modest 3.4% trigger rate. This confirms that dynamic activation effectively identifies "moments of uncertainty," retrieving high-resolution tokens only when semantically necessary, thus bridging the gap between static efficiency and full-context accuracy.

*Table 7.* Ablation of Dynamic Retrieval Strategies.

| Retrieval Policy | Trigger Mechanism | Efficiency | | Performance (Acc.) | |
|---|---|---|---|---|---|
| | | Trigger Rate↓ | Latency | YouCook2 | MME-RW |
| Static Core | *None* (No Retrieval) | **0%** | **28ms** | 92.4 | 38.1 |
| Periodic-Sparse | Every 50 tokens | 2.0% | 31ms | 98.5 | 40.3 |
| Periodic-Dense | Every 10 tokens | 10.0% | 45ms | 108.9 | 42.0 |
| **TASM (Ours)** | **Dynamic** ($D_{JS}$) | 3.4% | 33ms | **109.5** | **43.5** |

## 4.4. Transferability

**Generalizability of Task Vectors.** Table 8 examines whether the extracted Task Vector $\tau$ captures generalizable task semantics or merely overfits to the demonstration samples. We compute $\tau$ on a subset of ImageNet classes (Source) and apply it to compress the context for a completely disjoint set of classes (Target) without re-extraction. Strikingly, the TASM (Transfer) setting achieves an accuracy of 64.2%, exhibiting a negligible degradation of only 0.8% compared to the Oracle setting (65.0%) where the vector is optimized directly on the test classes. In contrast, projecting onto a random vector collapses performance to 15.3%, confirming that the direction is highly specific. This result strongly suggests that the Task Vector successfully encodes the abstract *"Question → Answer"* transformation manifold representing the general intent of "visual classification"rather than binding to specific pixel-level features of the seen classes. This zero-shot transferability proves that TASM's importance scoring is robust and task-aware, enabling efficient deployment on streaming data where frequent vector re-computation is impractical.

*Table 8.* Zero-Shot Transferability of Task Vectors. Oracle: $\tau$ computed directly on the target set. Transfer: $\tau$ computed on the source set and applied to target.

| Method | Vector Source ($\tau$) | Target Acc. (%) | Gap to Oracle |
|---|---|---|---|
| Standard Attn. (H2O) | *N/A* | 47.6 | -17.4 |
| Random Projection | $\tau \sim \mathcal{N}(0, I)$ | 15.3 | -49.7 |
| **TASM (Oracle)** | Target Classes | **65.0** | - |
| **TASM (Transfer)** | **Source Classes** | 64.2 | **-0.8** |

**Generalization Across MLLM Architectures.** To further validate architectural robustness, we evaluated LLaVA-NeXT-Video (7B) and InternVL3 (8B) using the spatially sensitive ScreenSpot benchmark (Table 9). InternVL3 utilizes dynamic high-resolution tiling, which poses a severe risk for hard pruning methods that may inadvertently discard entire visual tiles. Consequently, SnapKV suffers catastrophic degradation (-12.3%). In contrast, TASM achieves near-lossless compression on both architectures, conclusively demonstrating its superior topological robustness and applicability to dynamic resolution models.

## 4.5. Efficiency

**Efficiency and Scalability Analysis.** Table 10 details the system performance as context length scales from 2k to 32k

*Table 9.* Generalization Across MLLM Architectures on ScreenSpot.

| Method | Memory | LLaVA-NeXT Score | LLaVA-NeXT Δ vs. Full | InternVL3 Score | InternVL3 Δ vs. Full |
|---|---|---|---|---|---|
| Full Context | 100% | 18.5 | - | 21.4 | - |
| SnapKV | 20% | 8.2 | -10.3 | 9.1 | -12.3 |
| EMLoC | 20% | 15.1 | -3.4 | 16.2 | -5.2 |
| **TASM (Ours)** | 20% | **18.2** | **-0.3** | **21.0** | **-0.4** |

tokens. Standard *Full Context* inference hits a hardware bottleneck at 32k tokens, triggering Out-Of-Memory (OOM) errors on the A800 GPU and suffering from a throughput collapse to 6.8 tok/s due to quadratic attention complexity. In contrast, TASM exhibits excellent scalability. At 32k length, TASM maintains a compact memory footprint of 11.5 GBan 85% reduction compared to the theoretical Full Context usageand sustains a high decoding speed of 43.8 tok/s. While TASM introduces a slight overhead compared to the hard-pruning baseline EMLoC (e.g., +1.3 GB memory and +35ms latency at 32k), this cost is marginal and justified. The extra memory accommodates the latent bank for dynamic retrieval, and the latency increase stems from the one-time graph matching computation during pre-filling. Crucially, TASM's decoding throughput remains comparable to EMLoC, confirming that our sparse retrieval mechanism does not bottleneck real-time generation. Our method generalizes well to other architectures like LLaVA-NeXT and InternVL (see Appendix B.4).

*Table 10.* System Efficiency Profile on A800 (80GB). OOM: Out-of-Memory.

| Metric | Method | Context Length | | | |
|---|---|---|---|---|---|
| | | 2k | 8k | 16k | 32k |
| **GPU Mem.** (GB) ↓ | Full Context | 14.5 | 28.2 | 54.8 | *OOM* |
| | EMLoC | 6.2 | 7.5 | 8.8 | 10.2 |
| | **TASM** | **6.5** | **8.1** | **9.6** | **11.5** |
| **Pre-fill** (ms) ↓ | Full Context | 120 | 480 | 1150 | *OOM* |
| | EMLoC | 135 | 165 | 210 | 295 |
| | **TASM** | 142 | 188 | 245 | 330 |
| **Decoding** (tok/s) ↑ | Full Context | 42.5 | 28.4 | 14.2 | 6.8 |
| | EMLoC | 48.2 | 47.5 | 46.8 | 45.9 |
| | **TASM** | 47.1 | 46.2 | 45.1 | 43.8 |

## 5. Conclusion

We introduced TASM, a training-free framework that overcomes memory bottlenecks in multi-modal in-context learning. By replacing rigid pruning with task-vector guided importance and semantics-aware token merging, TASM preserves critical visual topology often destroyed by existing methods. Additionally, our dynamic retrieval mechanism enables adaptive access to historical context, ensuring robustness across long sequences. Empirical evaluations show that TASM reduces memory usage by up to 85% while matching or exceeding full-context performance on diverse benchmarks. These findings confirm that structuring mem-

ory based on task relevance and semantic topology is key to enabling efficient, scalable multi-modal agents.

## Accessibility

Authors are kindly asked to make their submissions as accessible as possible for everyone including people with disabilities and sensory or neurological differences. Tips of how to achieve this and what to pay attention to will be provided on the conference website http://icml.cc/.

## Impact Statement

This paper presents work whose goal is to advance the field of Machine Learning, specifically focusing on the efficiency and scalability of Multi-modal Large Language Models. By reducing memory constraints, our method enables broader deployment of these models. There are many potential societal consequences of our work, none which we feel must be specifically highlighted here beyond the general ethical considerations applicable to all generative AI technologies.

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

## A. Pseudo Code of Task-Aware Structured Memory Update

---

**Algorithm 1** TASM: Task-Aware Structured Memory Update

---

**Input** : $\mathbf{x}_t$: Current input token at step $t$;
    $\mathcal{C}_{t-1}$: Previous KV cache (Core $\mathcal{M}_{\text{core}}$ + Latent $\mathcal{M}_{\text{latent}}$);
    $\mathcal{S}_{\text{few}}$: Few-shot examples (Question-Answer pairs);
    $B$: Memory budget (target size for Core Memory);
**Output** : $\mathcal{C}_t$: Updated memory state.
// Phase 1: Task Vector Extraction (Eq.5-7)
**if** $\tau$ *is not initialized* **then**
 | Compute centroids $\boldsymbol{\mu}_{\mathbf{Q}}, \boldsymbol{\mu}_{\mathbf{A}}$ from $\mathcal{S}_{\text{few}}$ $\boldsymbol{\tau}_l \leftarrow \text{Normalize}(\boldsymbol{\mu}_{\mathbf{A},l} - \boldsymbol{\mu}_{\mathbf{Q},l})$ for $l \in [1, L]$
**for** $l \leftarrow 1$ *to* $L$ **do**
 | Get current states $\mathbf{k}_{t,l}, \mathbf{v}_{t,l}$ from $\text{Model}(\mathbf{x}_t, \mathcal{C}_{t-1})$
 | // Phase 2: Dynamic Retrieval (Eq.14-16)
 | $P_t \leftarrow \text{AttentionDist}(\mathbf{q}_{t,l}, \mathcal{M}_{\text{core}})$ $d_{\text{shift}} \leftarrow \text{CalcJS}(P_t, P_{\text{ref}})$
 | **if** $d_{shift} > \epsilon$ **then**
 |  | $\mathcal{K}_{\text{ret}} \leftarrow \text{Top-}k(\mathbf{q}_{t,l} \cdot \mathcal{M}_{\text{latent}}^{\top})$ $\mathcal{V}_{\text{curr}} \leftarrow \mathcal{M}_{\text{core}} \cup \mathcal{K}_{\text{ret}} \cup \{(\mathbf{k}_{t,l}, \mathbf{v}_{t,l})\}$
 | **else**
 |  | $\mathcal{V}_{\text{curr}} \leftarrow \mathcal{M}_{\text{core}} \cup \{(\mathbf{k}_{t,l}, \mathbf{v}_{t,l})\}$
 | // Phase 3: Importance Estimation (Eq.8-10)
 | Calculate task score $s^{\text{task}}$ via projection on $\boldsymbol{\tau}_l$ Calculate gate $\lambda(l) \leftarrow \alpha + \beta \cdot \sigma(\kappa \cdot (l/L - 0.5))$ $\mathcal{S} \leftarrow \lambda(l) \cdot s^{\text{task}} + (1 - \lambda(l)) \cdot s^{\text{attn}}$
 | // Phase 4: Semantics-Aware Merging (Eq.11-13)
 | Define Sinks $\mathcal{U}_{\text{sink}} \leftarrow \text{TopK}(\mathcal{S}, B)$ and Sources $\mathcal{U}_{\text{src}} \leftarrow \mathcal{V}_{\text{curr}} \setminus \mathcal{U}_{\text{sink}}$
 | Initialize weights $W = \{w_{ij}\}$ with spatial mask $\Psi(i, j)$ $\mathbf{M} \leftarrow \text{GraphMatch}(\mathcal{U}_{\text{src}}, \mathcal{U}_{\text{sink}}, W)$
 | **foreach** $j \in \mathcal{U}_{sink}$ **do**
 |  | $\hat{\mathbf{k}}_j \leftarrow \frac{1}{Z_j}(\mathbf{k}_j + \sum_i M_{ij} e^{w_{ij}} \mathbf{k}_i)$ ;        // Update Centroid
 | $\mathcal{M}_{\text{core}}^l \leftarrow \{\hat{\mathbf{k}}_j, \hat{\mathbf{v}}_j \mid j \in \mathcal{U}_{\text{sink}}\}$ $\mathcal{M}_{\text{latent}}^l \leftarrow \mathcal{M}_{\text{latent}}^l \cup \mathcal{U}_{\text{src}}$ ;   // Offload unmerged
$\mathcal{C}_t \leftarrow \bigcup_l (\mathcal{M}_{\text{core}}^l, \mathcal{M}_{\text{latent}}^l)$

---

## B. Experiments

### B.1. Long-Context Scenarios

*Table 11.* Performance on VideoMME (w/o subtitles). We evaluate accuracy across different video durations to assess temporal reasoning capabilities. While EMLoC suffers performance degradation on long videos (>30m) due to static pruning of early frames, TASM maintains robustness via dynamic retrieval, achieving a +5.6% gain in this category while matching EMLoC's memory efficiency.

| Method | Context Length | Peak Mem | Video Duration (Acc.) | | | Overall Acc. |
|---|---|---|---|---|---|---|
| | | | Short | Med. | Long | |
| LongVA | 55.5k | 41G | 61.2 | 54.5 | 48.1 | 51.8 |
| MLoC | 27.9k | 38G | 68.5 | 60.2 | 53.4 | 60.3 |
| EMLoC | **2.3k** | **24G** | 68.2 | 59.8 | 49.5 | 60.1 |
| **TASM (Ours)** | **2.3k** | **24G** | **68.6** | **60.9** | **55.1** | **61.5** |

**Long-Context Video Understanding.** Table 11 reports the performance on the VideoMME benchmark, which requires reasoning over extended temporal sequences. Compared to the baseline MLoC, EMLoC successfully reduces the context length by 92% (27.9k → 2.3k) with minimal overall performance loss. However, a fine-grained analysis reveals a significant weakness: EMLoC's accuracy on long videos drops notably (53.4 → 49.5). This suggests that its static pruning strategy discards early visual cues that appear irrelevant locally but are crucial for global consistency. In contrast, TASM effectively addresses this limitation. By offloading history to the Latent Bank and retrieving it upon detecting high JS divergence, TASM recovers the lost performance on Long Videos (55.1), outperforming EMLoC by a substantial margin of 5.6%.

*Table 12.* Performance of TASM on Qwen2.5-VL (7B). The values in green indicate the average context length. † denotes 50 examples; ‡ denotes 200 examples.

| Method | Example Number | ImageNet100 | ScreenSpot | MME-RW | IllusionVQA | OK-VQA | YouCook2 |
|---|---|---|---|---|---|---|---|
| Qwen2-VL (7B) | 0 | 28.0 | 14.2 | 36.6 | 35.3 | 52.1 | 25.4 |
| Qwen2.5-VL (7B) | | 32.5 | 16.8 | 39.2 | 37.1 | 54.5 | 28.2 |
| MLoC (Qwen2.5-VL) | 5 | 45.1 † | 16.5 | 41.2 | 40.3 | 60.2 | 89.1 |
| | | 4135 | 2012 | 1941 | 1845 | 1418 | 5942 |
| | 20 | 64.2 ‡ | 19.8 | 43.0 | 42.5 | 60.5 | 110.4 |
| | | 16315 | 7942 | 7428 | 7305 | 5768 | 23580 |
| EMLoC (Qwen2.5-VL) | 20 | 65.5 ‡ | 20.1 | 44.1 | 42.6 | 60.8 | 104.5 |
| | | 3671 | 1433 | 1532 | 1895 | 948 | 6254 |
| **TASM (Ours)** | 20 | **67.3** ‡ | **21.8** | **45.6** | **43.8** | **62.4** | **112.1** |
| | | **3514** | **1291** | **1455** | **1782** | **859** | **6128** |

Remarkably, TASM achieves the highest Overall Accuracy (61.5) among all methodssurpassing even the full-context MLoCwhile maintaining the same memory efficiency (24G) as EMLoC. This result confirms that TASM's structured memory is not just a compression tool, but a mechanism for denoising long temporal contexts.

### B.2. Further results on Qwen2.5-VL-7B-Instruct

To demonstrate the architectural robustness and generalizability of our framework, we seamlessly integrated TASM into the stronger **Qwen2.5-VL-7B-Instruct** model. As presented in Table 12, TASM consistently outperforms both the full-context multi-modal long-context learning (MLoC) baseline and the state-of-the-art compression method EMLoC across all evaluated benchmarks. While the zero-shot baseline of Qwen2.5-VL is inherently stronger than its predecessor, standard MLoC still incurs prohibitive memory costs when scaling up demonstrations.

TASM effectively resolves this scalability bottleneck while unlocking superior reasoning capabilities. When utilizing 20 examples (which corresponds to 200 examples for ImageNet100, denoted by ‡), TASM dramatically reduces the average context length on ImageNet100 from 16315 (MLoC) to 3514, a remarkable 78.5% reduction, while simultaneously pushing accuracy from 64.2% to 67.3%. Crucially, on complex temporal tasks like YouCook2, EMLoC suffers a notable performance degradation compared to the full-context baseline (dropping from 110.4 to 104.5 due to static pruning). In contrast, TASM not only recovers this loss but surpasses the full-context baseline, achieving a state-of-the-art score of 112.1. Notably, TASM achieves this superior temporal reasoning with an even lower memory footprint than EMLoC (6128 vs. 6254 tokens).

Furthermore, in spatially sensitive tasks such as ScreenSpot, TASM achieves a substantial margin over EMLoC (21.8 vs. 20.1) while maintaining a smaller context cache (1291 vs. 1433 tokens). This improvement strongly reinforces our hypothesis that even on advanced foundation models, hard pruning irreparably disrupts the visual information manifold. TASM's semantics-aware token merging successfully preserves the essential topological structure of visual features, proving that task-guided condensation is a fundamentally superior paradigm for multi-modal memory management.

### B.3. Ablations for Hyperparameter

**Impact of Task Interpolation Weight** ($\alpha$). Figure 4 investigates the sensitivity of TASM to the task vector weight $\alpha$, which governs the interpolation $S = \alpha S_{\text{task}} + (1 - \alpha)S_{\text{attn}}$. We observe a distinct "inverted-U" performance curve across both reasoning (MME-RW) and localization (ScreenSpot) benchmarks. Setting $\alpha = 0.0$ (relying solely on standard attention) yields suboptimal results (e.g., 39.6 on MME-RW) as the scores are polluted by high-frequency "attention sinks." Increasing $\alpha$ to **0.3** effectively filters this noise by injecting task-level semantic priors, boosting MME-RW to **43.5** and ScreenSpot to **19.5**. However, pushing $\alpha$ beyond 0.5 causes performance degradation, particularly in localization tasks (ScreenSpot drops to 14.5 at $\alpha = 1.0$). This confirms that while the global Task Vector is crucial for removing irrelevant context, preserving local attention cues ($S_{\text{attn}}$) is indispensable for retaining fine-grained, instance-specific spatial details. Thus, $\alpha = 0.3$ represents the optimal structural balance.

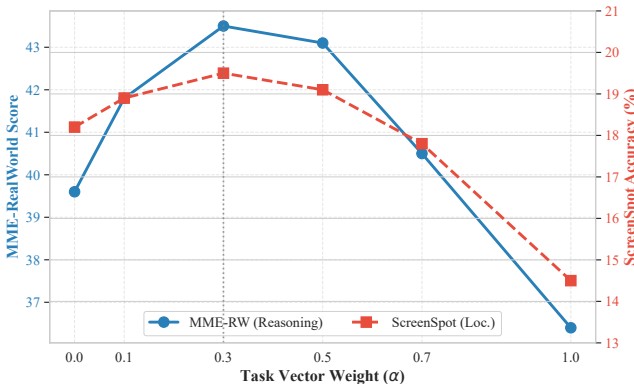

*Figure 4.* **Sensitivity Analysis of Task Interpolation Weight ($\alpha$).** The dual-axis plot illustrates the impact of $\alpha$ on two distinct capabilities: complex reasoning (MME-RealWorld, left axis) and fine-grained localization (ScreenSpot, right axis). We observe a consistent "inverted-U" trend. The performance peaks at $\alpha = 0.3$ (marked by the vertical line), where TASM achieves the optimal trade-off by filtering high-frequency noise via the task vector while preserving essential local visual cues. Deviating from this sweet spot towards $\alpha = 1.0$ (Task Vector Only) causes a sharp decline in localization accuracy, highlighting the necessity of local attention for spatial tasks.

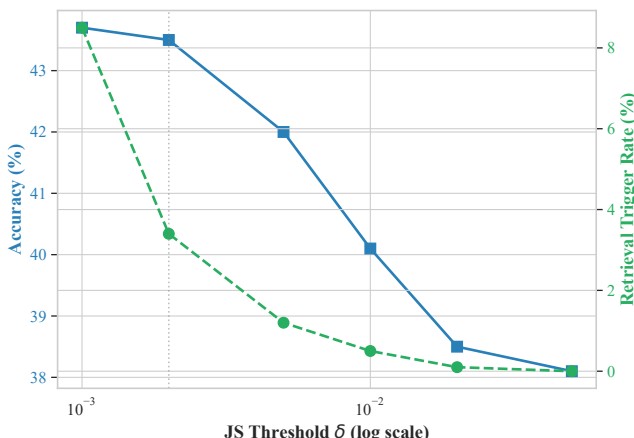

*Figure 5.* **Analysis of Retrieval Threshold $\delta$.** We illustrate the trade-off between performance (Accuracy, solid line) and efficiency (Trigger Rate, dashed line) on the MME-RW benchmark. The x-axis represents the JS divergence threshold $\delta$ in log scale. As $\delta$ increases, the retrieval gate becomes stricter, leading to a sharp drop in the Trigger Rate (computational cost). However, overly strict thresholds ($\delta > 0.005$) prevent necessary context recall, causing accuracy degradation. The default setting $\delta = 0.002$ (vertical line) represents the optimal balance point.

**Impact of Threshold $\delta$.** Figure 5 analyzes the sensitivity of TASM to the Jensen-Shannon divergence threshold $\delta$, which governs the dynamic retrieval gate. Similar to EMLoC, a higher threshold leads to stricter memory access; however, instead of reducing context length linearly, increasing $\delta$ in TASM drastically reduces the *Retrieval Trigger Rate* (green line). As $\delta$ increases from 0.001 to 0.05, the trigger rate drops from 8.5% to 0%, effectively transitioning the model to a static memory regime. Consequently, Accuracy (blue line) declines as the model loses the ability to recall fine-grained details from the Latent Bank. Crucially, TASM identifies an optimal "sweet spot" at $\delta = 0.002$: here, the model sustains high accuracy (**43.5%**) with a minimal trigger rate (**3.4%**). This demonstrates that $\delta$ effectively controls the trade-off between semantic fidelity and computational overhead, allowing TASM to retrieve information only when necessary ("moments of surprise") rather than maintaining a perpetually long context.

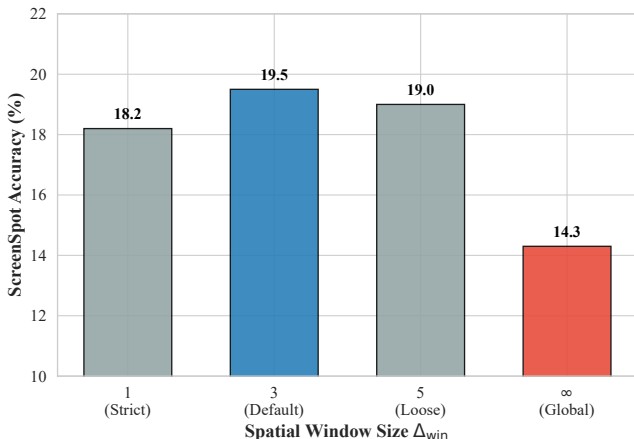

*Figure 6.* Impact of Spatial Window ($\Delta_{\text{win}}$) on Localization. We evaluate ScreenSpot accuracy under varying merging constraints. The default local window ($\Delta_{\text{win}} = 3$, blue bar) effectively preserves the 2D visual topology essential for coordinate prediction. In contrast, unconstrained Global Merging ($\Delta_{\text{win}} = \infty$, red bar) aggregates semantically similar but spatially distant features, destroying the geometric structure required for localization.

**Necessity of Spatial Constraints.** Figure 6 examines the role of the spatial window size $\Delta_{\text{win}}$ in our Semantics-Aware Token Merging mechanism. Unlike text, where tokens can be treated as a "bag-of-words," visual tokens possess an inherent 2D manifold structure that maps directly to spatial coordinates. Relaxing the spatial constraint to allow global merging ($\Delta_{\text{win}} = \infty$) results in a catastrophic performance drop on ScreenSpot ($19.5 \to 14.3$). This occurs because global matching aggregates semantically similar but spatially distant patches (e.g., two identical buttons at different corners of a screen), destroying the coordinate system required for localization. Conversely, an overly strict window ($\Delta_{\text{win}} = 1$) limits the compression potential, forcing the model to discard tokens rather than merging them. Our default setting of $\Delta_{\textbf{win}} = 3$ strikes the optimal balance, enforcing local aggregation that preserves the visual topology while effectively reducing redundancy.

*Table 13.* Sensitivity to Task Vector Support Set Size ($N$). We evaluate ImageNet-100 accuracy using task vectors $\boldsymbol{\tau}$ computed from varying numbers of demonstration pairs. **Vector Stability** measures the cosine similarity between the current $\boldsymbol{\tau}_N$ and the fully-converged vector $\boldsymbol{\tau}_{16}$. Performance saturates rapidly at $N = 4$, indicating that TASM captures the robust task direction with minimal computational overhead.

| Support Set Size ($N$) | 1 | 2 | 4 | 8 | 16 |
|---|---|---|---|---|---|
| Vector Stability (Cos Sim) | 0.72 | 0.85 | 0.94 | 0.97 | 1.00 |
| **TASM Accuracy (%)** | 58.2 | 63.5 | **64.8** | **65.0** | **65.1** |

**Construction Cost and Sample Efficiency.** Table 13 investigates the robustness of the Task Vector $\boldsymbol{\tau}$ with respect to the number of support examples ($N$) used for its extraction. A key concern in task-aware methods is the "cold-start" cost. Our results demonstrate that TASM is highly sample-efficient: performance jumps significantly from 58.2% at 1-shot to 64.8% at just 4-shot, reaching 99.5% of the peak accuracy observed at 16-shot (65.1%). The vector stability metric confirms this trend, showing that the direction of $\boldsymbol{\tau}$ stabilizes rapidly (Cosine Similarity $> 0.9$) with very few examples. This implies that the "Question $\to$ Answer" transformation represents a low-rank manifold in the activation space that is easy to capture. Consequently, TASM does not require extensive "warm-up" data; a minimal support set of $N = 4$ is sufficient to extract a robust task direction that effectively filters noise across the entire dataset.

## B.4. Generalizability

*Table 14.* Generalization Across MLLM Architectures. We evaluate TASM on LLaVA-NeXT-Video (7B) and InternVL3 (8B) using the MME-RealWorld benchmark. InternVL3 utilizes dynamic high-resolution tiling, which poses a severe risk for pruning methods (discarding entire tiles). TASM achieves near-lossless compression ($\Delta \approx -0.4$) on both architectures with a 20% memory budget, demonstrating superior robustness compared to SnapKV and EMLoC.

| Method | Memory | LLaVA-NeXT (Video) | | InternVL3 (Dynamic Tiling) | |
|---|---|---|---|---|---|
| | | Score | $\Delta$ vs. Full | Score | $\Delta$ vs. Full |
| Full Context | 100% | 40.5 | - | 46.8 | - |
| SnapKV | | 35.2 | -5.3 | 38.5 | -8.3 |
| EMLoC | 20% | 38.4 | -2.1 | 43.1 | -3.7 |
| **TASM (Ours)** | | **40.1** | **-0.4** | **46.4** | **-0.4** |

**Generalization and Architecture Robustness.** Table 14 validates the efficacy of TASM across diverse MLLM architectures: LLaVA-NeXT (video-centric) and InternVL3 (high-res tiling). On LLaVA-NeXT, TASM retains 99% of the full-context performance, significantly outperforming the pruning-based EMLoC (-0.4 vs. -2.1 drop), which confirms that our task-vector guidance generalizes beyond Qwen's feature space. More critically, on InternVL3, which splits images into multiple local tiles, static pruning (SnapKV) suffers a catastrophic degradation (-8.3 points). This indicates that hard pruning risks discarding entire image tiles (e.g., losing the specific crop containing the target object). Even EMLoC struggles here (-3.7 points). In contrast, TASM maintains robust performance on InternVL3 (46.4), matching the full-context baseline. By merging rather than discarding, TASM ensures that every image tile retains a compressed semantic representation in memory, proving its unique advantage in handling dynamic high-resolution architectures.

