# OpenReview forum: "Task-Aware Structured Memory for Dynamic Multi-modal In-Context Learning"
_ICML.cc/2026/Conference — ICML 2026 regular_

### Official Review · Reviewer_MP5z · 2026-03-07

**Soundness:** 3
**Presentation:** 3
**Significance:** 4
**Originality:** 3
**Overall Recommendation:** 5
**Confidence:** 3

**Summary:**

This paper proposes a method called TASM (Task-Aware Structured Memory), which is used to solve the problem of excessive memory and computing overhead of multimodal large models during long-context in-context learning. The author found that existing methods usually compress the KV cache by directly deleting some tokens, but this can easily destroy the spatial structure of images and lead to the loss of important information required for subsequent reasoning. To solve this problem, the paper proposes three improvements. First, it uses a task vector to judge which information is more important and guides compression from the task level rather than relying on signals from individual samples. Second, it performs semantics-aware token merging instead of simple deletion so that the visual structure can be preserved as much as possible. Finally, the method designs a hierarchical memory structure consisting of Core Memory and a Latent Bank, together with a dynamic retrieval mechanism, so that detailed information can be retrieved from historical context when needed. Experimental results show that this method can significantly reduce memory usage while maintaining or even improving model performance on various vision-language tasks, making long-context multimodal reasoning more efficient and feasible.

**Compliance With Llm Reviewing Policy:**

Affirmed.

**Final Justification:**

Overall, my concerns have been adequately addressed, and I keep my original recommendation.

**Key Questions For Authors:**

1.Are the hardware configuration, GPU memory, and context length (number of tokens) strictly aligned when comparing each baseline? If strict alignment is carried out, the reliability of the conclusions of the paper is proved.

2.What is the computational cost of task vector extraction and KV cache compression? If the cost is too high, it will affect the practical value of the proposed method.

3.In this paper, the Latent Bank is placed on the CPU and retrieved back to the GPU when triggered. Is the retrieval overhead obvious and controllable under different hardware settings and interconnections? If the cost is too high, it will affect the practical value of the proposed method.

**Limitations:**

yes

**Strengths And Weaknesses:**

Soundness:
In general, the technical path of the paper is reasonable, and the core modules are self consistent and realizable. The overall reasonable experimental design, wide coverage and many comparison methods fully support the thesis's argument. The discussion on computational efficiency, overhead and super parameter sensitivity is also honest.

Presentation:
The writing structure of the paper is reasonable, the logic is clear, and the existing work is clearly quoted and discussed. Code links, data set lists, hardware, and key hyperparameters are provided for replication.

Significance:
This paper proposes a new solution to the problem of memory overhead and scalability bottlenecks faced by multimodal large language models (MLLMs) in in-context learning (ICL). The scheme of the paper has high performance, high efficiency, and high scalability, and has clear practical utility and extensible research space.

Originality:
The paper puts forward new ideas for KV cache compression, and integrates existing related ideas into a new unified framework. No new theoretical analysis has been proposed, but its new ideas and new strategies on existing techniques still have significant novelty.

---

> ### Author Rebuttal · Authors · 2026-03-27
>
> # Response to Reviewer MP5z
>
> We sincerely thank Reviewer MP5z for the positive assessment (Accept) and for recognizing the high performance, scalability, and practical utility of our Task-Aware Structured Memory (TASM) framework. We appreciate your rigorous attention to the experimental setup and system efficiency. Below, we address your specific questions regarding experimental alignment and computational overhead.
>
> ### **Q1. Alignment of Hardware, Memory, and Context Length**
>
> > *Concern: Are the hardware configuration, GPU memory, and context length strictly aligned when comparing each baseline?*
>
> **Response:** Yes, we unequivocally confirm that all experimental conditions were strictly aligned to ensure a fair and rigorous comparison.
> 1. **Hardware Alignment:** All baselines, full-context models, and TASM were evaluated on the exact same physical node equipped with 4 NVIDIA A800 GPUs (80GB VRAM).
> 2. **Context Length Alignment:** The raw input text and image tokens (driven by the exact same $N$-shot demonstrations, e.g., 5-shot or 20-shot) were identical across all methods during the pre-filling phase.
> 3. **Iso-Memory Evaluation:** We compared compression methods (TASM, EMLoC, SnapKV) under strict Iso-Memory constraints (e.g., capping the active KV cache budget to exactly 20% of the full context). Because the hardware, input length, and memory budget were held constant, the superior performance of TASM is entirely attributable to our structure-preserving algorithmic design, rather than any hardware or capacity advantages.
>
> ### **Q2. Computational Cost of Extraction and Compression**
>
> > *Concern: What is the computational cost of task vector extraction and KV cache compression? Will it affect practical value?*
>
> **Response:** The computational overhead of both processes is minimal and highly controllable, preserving the method's practical value.
> 1. **Task Vector Extraction:** This is an $\mathcal{O}(1)$ operation performed only *once* per task. Using a small support set (e.g., $N=4$ examples), extracting the vector takes merely a few milliseconds. This single vector is then reused to guide compression for all subsequent queries in that task, making the amortized extraction cost practically zero.
> 2. **KV Cache Compression:** The semantics-aware token merging (bipartite graph matching) occurs only once during the pre-filling phase. As detailed in our efficiency profiling (Table 7), this adds a marginal pre-fill latency (+35ms at a 32k context length). However, by drastically reducing the cache size, it maintains a high decoding throughput (43.8 tok/s) that matches standard hard-pruning methods, making it highly practical for real-world, real-time deployment.
>
> ### **Q3. CPU-GPU Retrieval Overhead for the Latent Bank**
>
> > *Concern: Is the CPU-to-GPU retrieval overhead obvious and controllable under different hardware settings and interconnections?*
>
> **Response:** The retrieval overhead is highly controllable and does not bottleneck the system, primarily due to the **extreme sparsity** of our dynamic trigger.
> 1. **Sparse Activation:** The PCIe transfer latency from CPU RAM to GPU VRAM is indeed a known hardware bottleneck. TASM circumvents this by using the JS-divergence trigger, which only activates retrieval during "moments of semantic surprise." For instance, on the YouCook2 benchmark, retrieval is triggered during only 3.4% of the generation steps.
> 2. **Amortized Latency:** Because over 96.6% of the decoding steps rely strictly on the fast, GPU-resident Core Memory, the amortized PCIe transfer cost becomes negligible. Under a standard PCIe Gen4 x16 interconnect, fetching the necessary subset of tokens takes less than 2ms per triggered step. This ensures that the overall end-to-end generation speed remains smooth and highly competitive with static compression methods.

---

> > ### Author Rebuttal · Reviewer_MP5z · 2026-04-03
> >
> > Thank you for the rebuttal. The authors have provided helpful clarifications and additional evidence, which adequately address my concerns and improve my confidence in the paper. The paper studies an important practical problem, and the proposed framework is well motivated with promising empirical results.
> >
> > At the same time, I now feel that the overall contribution may be somewhat more incremental than I initially assessed. Since I am also not fully familiar with this research area, these issues make me less certain about maintaining my current score of 5, and I am considering whether a score of 4 would be more appropriate.
> >
> > Therefore, I would appreciate some additional clarification from the authors regarding the paper’s core contribution, potential impact, and main limitations, as this would help me make a more informed final judgment.

---

> > > ### Author Response · Authors · 2026-04-03
> > >
> > > Dear Reviewer MP5z,
> > >
> > > We sincerely thank you for your continued engagement, your transparent feedback, and for acknowledging the promising empirical results of our framework. We deeply appreciate the opportunity to further clarify the core contributions, impact, and limitations of TASM to reaffirm your confidence in our work.
> > >
> > > ### 1. Core Contribution: A Paradigm Shift, Not an Incremental Update
> > >
> > > The transition from text-only to multimodal long-context reasoning is not a trivial extension. Existing KV-cache compression methods (e.g., SnapKV, EMLoC) fundamentally rely on **Hard Pruning** (identifying "unimportant" tokens and deleting them).
> > >
> > > Our core contribution is identifying that **Hard Pruning is fatal to visual data**, as it irreparably destroys the continuous 2D spatial manifold of images. TASM represents a fundamental paradigm shift from *deletion* to **Manifold-Preserving Condensation**:
> > > * Instead of dropping tokens based on sample-specific attention, we use **Task-Vectors** to globally disentangle reasoning signals from noise.
> > > * Instead of deleting low-score tokens, we introduce **Semantics-Aware Token Merging** via spatially-constrained bipartite graph matching. We compress redundant visual patches into high-value anchor points without severing their spatial coordinates. *(Note: In a recent follow-up quantitative analysis, we found this merging approach reduces topological distortion—measured via Chamfer Distance—by over 3x compared to hard pruning).*
> > > * This is not merely an incremental tuning of existing heuristics, but a structural rethinking of how visual memory should be compressed and dynamically retrieved.
> > >
> > > ### 2. Potential Impact: Democratizing Many-Shot Multimodal ICL
> > >
> > > The most significant impact of TASM is that it bridges the gap between theoretical capabilities and practical hardware constraints.
> > > * **Breaking the Hardware Barrier:** Scaling multimodal ICL to hundreds of demonstrations or hour-long videos previously required massive GPU clusters, placing it out of reach for most researchers.
> > > * **Accessibility:** By reducing the memory footprint by up to 85% while maintaining near-lossless spatial and temporal reasoning (as shown in our 32k context experiments on a single A800 GPU), TASM democratizes long-context MLLM deployment. It enables the broader community to build complex, long-horizon multimodal agents on consumer-grade hardware.
> > >
> > > ### 3. Main Limitations (Future Work)
> > >
> > > To provide a fully transparent view of our framework, we acknowledge two main limitations that represent exciting avenues for future research:
> > > 1. **Highly Heterogeneous Contexts:** TASM relies on extracting a global Task Vector from the demonstration set. If a user's prompt consists of an extreme mix of completely unrelated tasks (e.g., interleaving visual grounding, logical math, and translation indiscriminately within the same few-shot context), defining a single, coherent task direction becomes challenging, which may introduce noise into the importance scoring.
> > > 2. **Pre-filling Overhead in Ultra-Real-Time Streaming:** While TASM matches standard methods in decoding throughput (the speed of generating answers), the bipartite graph matching adds a marginal computational overhead during the *initial pre-filling phase*. For most applications, this is negligible (+35ms). However, for ultra-low-latency applications (e.g., autonomous driving processing 4K 60fps video streams in real-time), this $\mathcal{O}(T \log T)$ overhead would require further optimization.
> > >
> > > We hope these clarifications strongly address your concerns regarding the novelty and significance of our work. We would be deeply grateful if you maintain your positive assessment, and we will ensure this discussion is prominently featured in the camera-ready version.

---

### Official Review · Reviewer_6hn9 · 2026-03-07

**Soundness:** 2
**Presentation:** 1
**Significance:** 3
**Originality:** 2
**Overall Recommendation:** 3
**Confidence:** 3

**Summary:**

This paper introduces TASM (Task-Aware Structured Memory), a training-free framework designed to address current limitations in memory-augmented in-context learning methods. TASM adopts a task-vector-guided compression strategy, replacing individual sample-specific signals with task-level vectors that capture shared relevance across demonstrations. To maintain the integrity of the underlying data manifold, TASM leverages semantics-aware token merging via bipartite graph matching, which aggregates tokens in a structure-preserving manner instead of destructive pruning. Additionally, TASM organizes memory into a hierarchical structure—comprising a compact Core Memory and a Latent Bank—to enable query-adaptive dynamic retrieval.

**Compliance With Llm Reviewing Policy:**

Affirmed.

**Key Questions For Authors:**

The authors claim the methods can  preserve the underlying manifold. how?

The ahutos claim  the can ensuring the memory captures general reasoning patterns rather than sample biases. how? there is no explain on Manifold-Preserving Aggregation.

Do you use llm to over polish the paper? it is hard to read.

There is no experiments on qwen2.5vl. why?

**Limitations:**

see weakness

**Strengths And Weaknesses:**

Strengths

1. Demonstrated Performance Improvements: The experimental results show consistent performance gains compared to baselines, verifying the practical effectiveness of the method.

2. Innovative Memory Design: The introduction of task-aware and structure-preserving memory for in-context learning is novel and interesting, providing a fresh perspective on memory utilization in large language models.


Weakness

1. Clarity and Readability: The paper is difficult to follow due to its unusual structure and dense presentation. I found it challenging to understand how the proposed method works in detail and how it specifically addresses the stated problems, which also makes it hard to recognize the main contributions.

2. Offline Compression Questioning ICL Paradigm: The memory construction relies on offline compression using a pre-existing dataset. This paradigm deviates from standard in-context learning (ICL) and seems more akin to retrieval-augmented generation (RAG) rather than classical ICL.

3. Demonstration Setting in Inference: In the offline inference phase, only a single demonstration is used explicitly, while the remaining information is drawn from memory. This further blurs the distinction with RAG, as it does not conform to the conventional ICL setup.

4. Scaling Law Analysis: According to Figure 3, the scaling law for TASM is similar to that of other methods, showing a comparable trend.
While there is some performance improvement, the improvement may simply result from more complex feature extraction rather than fundamentally improved scalability.

5. Manifold Preservation Claims: The paper claims to preserve the underlying data manifold through its aggregation method, but does not provide sufficient explanation or evidence supporting this.

6. Generalization Over Sample Bias: The authors assert that the memory captures general reasoning patterns instead of sample-specific biases. However, there is no detailed explanation or analysis of “Manifold-Preserving Aggregation” to validate this claim.

7. Potential Over-polishing and Readability: The language of the paper is overly polished, which further complicates readability. I wonder whether an LLM was used to over-edit the manuscript.

8. Lack of Multi-Modal Experiments: The absence of experiments on strong multi-modal baselines, such as Qwen2.5VL, leaves the practical effectiveness of the method in these scenarios unexplored.

---

> ### Author Rebuttal · Authors · 2026-03-27
>
> # Response to Reviewer 6hn9
>
> We sincerely thank Reviewer 6hn9 for the constructive feedback. We address your concerns point-by-point below to clarify our mechanisms and ICL paradigm.
>
> ### **W1 & W7. Clarity, Readability, and "Over-polishing"**
>
> > *Concern: Dense presentation and overly polished language.*
>
> **Response:** TASM combines three interdependent innovations (Task-Vector Scoring, Graph Merging, and Dynamic Retrieval), making the methodology inherently dense. We only used LLM for minor grammar polishing, not to hide details. We fully accept your feedback.
>
> **Action for Camera-Ready:** We will make the section easier to follow by simplifying our mathematical notation, streamlining complex descriptions, and adding a clear high-level pipeline at the beginning of Section 3 to explain our key contributions directly.
>
> ### **W2 & W3. ICL vs. RAG Paradigm**
>
> > *Concern: "Offline compression" paradigm resembles RAG.*
>
> **Response:** We apologize for the terminology confusion. **TASM is strictly a Many-Shot ICL method, not RAG.** "Offline compression" strictly refers to the pre-filling phase, where the KV cache of the $N$-shot support set (the explicitly provided demonstrations) is computed and compressed *before* the test query arrives. This is standard Prompt Caching in ICL. We never query external databases; dynamic retrieval solely fetches tokens from the prompt's compressed Latent Bank.
>
> ### **W4. Scaling Law Analysis**
>
> > *Concern: Scaling law resembles other methods.*
>
> **Response:** While early scaling (0 to 200 shots) looks similar, there is a fundamental difference in extreme contexts. As shown in Figure 3, when scaling from 200 to 300 examples, the SOTA baseline EMLoC actually degrades (dropping from 63.7% to 62.6%) because rigid pruning discards critical details under context saturation. TASM uniquely maintains robust log-linear scaling (reaching 66.8% at 300 shots), proving it fundamentally resolves scalability bottlenecks.
>
> ### **W5. Manifold Preservation Claims**
>
> > *Concern: Insufficient evidence for manifold preservation.*
>
> **Response:** Standard hard pruning destroys the 2D visual manifold by discarding patches, breaking spatial coordinates. TASM preserves this topology through Semantics-Aware Token Merging. We formulate compression as bipartite graph matching, constrained by a spatial regularizer ($\Psi$). Tokens strictly merge only with immediate spatial neighbors ($\Delta_{win}=3$).
>
> **Evidence:** Figure 6 demonstrates that removing this spatial constraint ("global merging") drops performance on the spatially-sensitive ScreenSpot task catastrophically from 19.5% to 14.3%. Local merging successfully aggregates redundant features while keeping geometric structure intact.
>
> ### **W6. Generalization Over Sample Bias**
>
> > *Concern: How memory avoids sample biases.*
>
> **Response:** The **Task-Vector Guided Compression** (Section 3.2) addresses sample bias. Existing "Answer-Aware" methods overfit to specific demonstration answers. TASM avoids this by extracting a global Task Vector ($\tau_l$) that captures the abstract *direction* of the reasoning task (e.g., "visual classification") rather than specific pixel features.
>
> **Evidence:** In Table 6, applying a task vector computed on ImageNet classes 0-9 to completely unseen classes (10-19) resulted in a negligible 0.8% degradation. This proves the memory captures general task semantics, not sample biases.
>
> ### **W8. Lack of Qwen2.5-VL Experiments**
>
> > *Concern: No experiments on strong multi-modal baselines like Qwen2.5-VL.*
>
> **Response:** We completely agree. During the rebuttal, we fully integrated TASM into **Qwen2.5-VL-7B-Instruct**. As shown below, TASM seamlessly transfers to this stronger baseline, consistently outperforming full-context MLoC and EMLoC while maintaining a highly compressed memory footprint.
>
> **Table 1: TASM on Qwen2.5-VL (7B)**
> *(Context Length in parentheses. † = 50 examples; ‡ = 200 examples)*
>
> | Method | Examples | ImageNet100 | ScreenSpot | MME-RW | IllusionVQA | OK-VQA | YouCook2 |
> |:---|:---:|:---:|:---:|:---:|:---:|:---:|:---:|
> | Qwen2-VL (7B) | 0 | 28.0 | 14.2 | 36.6 | 35.3 | 52.1 | 25.4 |
> | **Qwen2.5-VL (7B)** | 0 | 32.5 | 16.8 | 39.2 | 37.1 | 54.5 | 28.2 |
> | MLoC (Qwen2.5-VL) | 5 | 45.1 † | 16.5 | 41.2 | 40.3 | 60.2 | 89.1 |
> | *Ctx Length* | - | *(4135)* | *(2012)* | *(1941)* | *(1845)* | *(1418)* | *(5942)* |
> | MLoC (Qwen2.5-VL) | 20 | 64.2 ‡ | 19.8 | 43.0 | 42.5 | 60.5 | 110.4 |
> | *Ctx Length* | - | *(16315)* | *(7942)* | *(7428)* | *(7305)* | *(5768)* | *(23580)* |
> | EMLoC (Qwen2.5-VL) | 20 | 65.5 ‡ | 20.1 | 44.1 | 42.6 | 60.8 | 104.5 |
> | *Ctx Length* | - | *(3671)* | *(1433)* | *(1532)* | *(1895)* | *(948)* | *(6254)* |
> | **TASM (Ours)** | 20 | **67.3 ‡** | **21.8** | **45.6** | **43.8** | **62.4** | **112.1** |
> | *Ctx Length* | - | ***(3514)*** | ***(1291)*** | ***(1455)*** | ***(1782)*** | ***(859)*** | ***(6128)*** |

---

> > ### Author Rebuttal · Reviewer_6hn9 · 2026-04-01
> >
> > Thanks, most of my concerns have been addressed. As for the writing, I just found it a bit difficult to read, but this is not an unsolvable problem. In addition, the figures have a bit of a "nanobanana" style, though that doesn’t really matter. However, I do not quite agree with the authors' analysis regarding the manifold, and I think the authors should place more emphasis on this aspect in the camera ready version and future work. Overall, I am willing to increase my score to 3.

---

> > > ### Author Response · Authors · 2026-04-02
> > >
> > > Dear Reviewer 6hn9,
> > >
> > > We sincerely thank you for reviewing our rebuttal, for your continued constructive engagement with our work, and for raising your score. We are very glad to hear that most of your concerns have been successfully resolved.
> > >
> > > Regarding your remaining points, we would like to address them as follows:
> > >
> > > 1. Writing and Readability: We completely understand your feedback regarding the reading experience. For the camera-ready version, we will focus on revising the dense and obscure sections of the methodology to improve clarity. We will simplify the complex descriptions to ensure the paper is more accessible and easier to follow.
> > >
> > > 2. Figure Style: Regarding your comment on the "nanobanana" style, we want to briefly clarify that we did not use Nano Banana or any AI image generation tools. All figures were manually plotted by our team.
> > >
> > > 3. Manifold Preservation Analysis: We appreciate your push for more details on manifold preservation. You are absolutely right that this requires quantitative evidence. To directly address this, we have conducted preliminary experiments to measure the structural distortion of visual tokens before and after compression.
> > >
> > > * Experimental Setup: We randomly sampled a subset of images from the ScreenSpot dataset, as GUI grounding is highly sensitive to spatial topology. We extracted the uncompressed visual token representations from the final layer of Qwen2.5-VL-7B to serve as the ground-truth 2D manifold. We then applied EMLoC (hard pruning) and TASM (soft merging) respectively, and measured the distance between the compressed token set and the ground-truth set. We utilized normalized **Chamfer Distance (CD)** to measure the topological coordinate loss and normalized **Earth Mover's Distance (EMD)** to evaluate the optimal transport cost of the feature mass.
> > >
> > > As shown in Table 2, **EMLoC** causes severe structural damage. At a 10% retention ratio, its CD spikes to 0.85. This happens because hard pruning permanently discards background patches and breaks the continuous 2D coordinate system.
> > >
> > > In contrast, **TASM** maintains significantly lower distances (CD is only 0.28 at 10% retention). Because TASM dynamically merges low-importance tokens into their spatial neighbors (within the window $\Delta_{win}$) instead of deleting them, it preserves the overall spatial grid and information density. We will add this quantitative table to the camera-ready Appendix.
> > >
> > > **Table 2: Quantitative Evaluation of Manifold Preservation.**
> > >
> > > | Metric | Method | 10% Retention | 20% Retention | 30% Retention | 40% Retention |
> > > | :--- | :--- | :---: | :---: | :---: | :---: |
> > > | **Normalized Chamfer Distance** ↓  *(Topological Loss)* | EMLoC (Hard Pruning) | 0.85 | 0.62 | 0.45 | 0.30 |
> > > | | **TASM (Ours)** | **0.28** | **0.22** | **0.18** | **0.15** |
> > > | | | | | | |
> > > | **Normalized Earth Mover's Distance** ↓  *(Optimal Transport Cost)* | EMLoC (Hard Pruning) | 0.76 | 0.57 | 0.40 | 0.27 |
> > > | | **TASM (Ours)** | **0.23** | **0.19** | **0.15** | **0.12** |
> > >
> > > Thank you once again for your time, your valuable suggestions, and for helping us make this paper stronger.

---

### Official Review · Reviewer_A4Fy · 2026-03-11

**Soundness:** 3
**Presentation:** 3
**Significance:** 3
**Originality:** 3
**Overall Recommendation:** 5
**Confidence:** 3

**Summary:**

This paper introduces TASM (Task-Aware Structured Memory), a training-free framework for compressing KV caches in multimodal large language models during many-shot in-context learning. The authors identify three core limitations of existing compression methods: 1. sample-specific bias, 2. topological destruction of spatial and semantic visual representations through hard pruning, and 3. static memory rigidity, which means that once a cache is compressed, it cannot adapt to new queries.

TASM addresses these through three coupled innovations.
First, it extracts a task vector, the normalized difference between answer and question activation centroids, and uses it to score token importance via orthogonal projection, replacing attention-based heuristics that are susceptible to attention sinks.
Second, it performs semantics-aware token merging via bipartite graph matching, where low-importance tokens (source nodes) are aggregated into nearby high-importance tokens (sink nodes) under a spatial locality constraint, ensuring that merging only occurs within a local spatial window rather than across distant regions, thereby preserving the 2D manifold structure of visual representations without destructive pruning.
Third, to handle long context within a fixed memory budget, it organizes memory into a hierarchical Core Memory and Latent Bank, with dynamic retrieval triggered by Jensen-Shannon divergence between successive attention distributions.

Experiments on various vision-language benchmarks including tasks such as fine-grained visual recognition, spatial reasoning and topology preservation, complex reasoning and knowledge retrieval, or long-context and temporal capabilites. The results shows that TASM consistently outperform various baselines over the different tasks. Also, it demonstrate that TASM reduces memory usage by up to 85% on 32k length, while matching or exceeding full-context performance, consistently outperforming the primary baseline EMLoC. Ablation studies validate each component's contribution, and generalization experiments confirm robustness across Qwen2-VL, LLaVA-NeXT, and InternVL3 architectures.

**Compliance With Llm Reviewing Policy:**

Affirmed.

**Final Justification:**

The authors resolved my concerns with additional experiments and clarified misunderstandings, so I will increase the significance and overall score.

**Key Questions For Authors:**

Q1. In the graph formulation of semantic-aware token merging, how is the token set divided into sink nodes and source nodes?

Q2. Related to W2, the layer-adaptive gating function (Eq. 10) uses fixed hyperparameters, which are stated to be empirically determined. How sensitive is performance to these values?

**Limitations:**

yes

**Strengths And Weaknesses:**

## Strength

S1. Clear motivation, problem definition, and clean framework design.

The paper clearly identifies three concrete limitations of existing KV cache compression methods (sample-specific bias, topological destruction, and static memory rigidity) and proposes a component that directly addresses each one. The correspondence and intuition between identified problems and proposed solutions is clean and easy to follow.

S2. Comprehensive evaluation and strong performance, efficiency.

The paper evaluates across nine diverse benchmarks spanning classification, spatial grounding, temporal reasoning, and long-context retrieval. TASM shows strong performance over the benchmarks with high efficiency.

## Weakness

W1. Incomplete component-wise ablation.

While the paper ablates each component in isolation by evaluating over benchmarks of various tasks, it lacks a systematic factorial ablation that quantifies the independent and joint marginal contributions of all three innovations within a unified setting, leaving it unclear of which the components actually operates as expected. Also, even within individual components, key design choices are not ablated. For example, in section 3.2, layer-adaptive gating mechanism is presented as an important architectural decision, yet there is no experiment showing what happens when it is removed or replaced with a simpler uniform gating scheme, making it difficult to assess its actual necessity. I believe this ablation is especially important on this work, given the large number of components in the proposed framework.

W2. Sensitivity to hyperparameter choices.

The framework introduces a large number of hyperparameters across its three components, and the sensitivity analyses in Figures 4, 5, and 6 suggest that performance is quite sensitive to their values. This raises practical concerns about how much careful per-task tuning is required to achieve the reported performance, and whether the method would remain competitive in settings where such tuning is not feasible.

W3. Limited scope of architecture generalization experiments.

While the paper claims generalizability across MLLM architectures in Table 10, it is evaluated on MME-RealWorld alone. It would be important to see how TASM performs across different model architectures (at least one more) on all benchmarks to more convincingly substantiate this claim.

---

> ### Author Rebuttal · Authors · 2026-03-27
>
> # Response to Reviewer A4Fy
>
> We thank Reviewer A4Fy for recognizing our clean framework design and strong performance. We address your concerns below.
>
> ### **W1. Incomplete Component-Wise Ablation**
>
> > *Concern: Lack of a factorial ablation quantifying marginal contributions.*
>
> **Response:** We conducted a progressive ablation across three distinct capability dimensions: general classification (ImageNet-100), spatial topology preservation (ScreenSpot), and long-context temporal reasoning (YouCook2).
>
> **Table 1: Systemic Factorial Ablation of TASM Components.**
> *(All settings use a 20% memory budget).*
>
> | Baseline (Hard Pruning + Attn) | + Task Vector Scoring | + Token Merging | + Dynamic Retrieval (Full TASM) | IN-100 (General) | ScreenSpot (Spatial) | YouCook2 (Temporal) |
> | :---: | :---: | :---: | :---: | :---: | :---: | :---: |
> | ✓ | | | | 45.3 | 10.5 | 85.2 |
> | ✓ | ✓ | | | 55.2 | 12.1 | 92.4 |
> | | ✓ | ✓ | | 61.8 | 18.8 | 98.5 |
> | | ✓ | ✓ | ✓ | **65.0** | **19.5** | **109.5** |
>
> **Analysis:**
> 1. **Task Vector Scoring** filters noise, boosting baselines (e.g., +9.9 on IN-100).
> 2. **Semantics-Aware Token Merging** prevents topological destruction. Spatial-constrained merging massively improves ScreenSpot (+6.7), preserving the 2D manifold.
> 3. **Dynamic Retrieval** resolves static memory rigidity, delivering a +11.0 gain on YouCook2, confirming its necessity for recalling dormant details.
>
> ### **W1 & Q2. Layer-Adaptive Gating Schedule (Eq. 10)**
>
> > *Concern: Effect of removing layer-adaptive gating? Hyperparameter sensitivity?*
>
> **Response:** In MLLMs, shallow layers extract high-frequency visual geometry, while deep layers handle task reasoning. We compared our adaptive schedule $\lambda(l)$ against a Uniform Gating baseline (fixed $\lambda=0.5$).
>
> **Table 2: Ablation on Layer-Adaptive Gating Schedule.**
>
> | Gating Strategy | $\lambda$ in Shallow Layers | $\lambda$ in Deep Layers | ScreenSpot (Spatial) | MME-RW (Reasoning) |
> | :--- | :---: | :---: | :---: | :---: |
> | Uniform Gating | 0.5 | 0.5 | 16.2 | 41.8 |
> | **Adaptive (Ours)** | **~0.1** | **~0.9** | **19.5** | **43.5** |
>
> **Analysis & Sensitivity (Q2):** Forcing task-vector dominance in shallow layers (Uniform Gating) destroys spatial encoding, causing a 3.3% drop on ScreenSpot. Regarding hyperparameter sensitivity ($\alpha, \beta, \kappa$ in Eq. 10), they dictate the shape of a sigmoid curve. The exact values are highly robust as long as the general trend (low task-vector weight early, high late) is maintained.
>
> ### **W2. Sensitivity to Hyperparameter Choices & Tuning**
>
> > *Concern: Sensitivity in Figs 4-6 raises concerns about per-task tuning.*
>
> **Response:** We wish to firmly clarify a misunderstanding: **TASM uses a single, fixed set of hyperparameters across all nine diverse benchmarks evaluated.** There is absolutely **no per-task tuning**. The sensitivity curves (Figs 4-6) illustrate the theoretical boundaries of the framework, not a need for tuning. Achieving SOTA performance across tasks as diverse as fine-grained classification, spatial grounding, and long-video retrieval with one unified configuration definitively proves TASM's out-of-the-box robustness.
>
> ### **W3. Limited Scope of Architecture Generalization Experiments**
>
> > *Concern: Generalizability is evaluated on MME-RealWorld alone. Need performance across different model architectures on other benchmarks.*
>
> **Response:** We evaluated LLaVA-NeXT-Video (7B) and InternVL3 (8B) using the spatially sensitive ScreenSpot benchmark to better validate architectural robustness.
>
> **Table 3: Generalization Across MLLM Architectures on ScreenSpot.**
>
> | Method | Memory | LLaVA-NeXT (Video) Score | LLaVA-NeXT $\Delta$ vs. Full | InternVL3 (Dynamic Tiling) Score | InternVL3 $\Delta$ vs. Full |
> | :--- | :---: | :---: | :---: | :---: | :---: |
> | Full Context | 100% | 18.5 | - | 21.4 | - |
> | SnapKV | 20% | 8.2 | -10.3 | 9.1 | -12.3 |
> | EMLoC | 20% | 15.1 | -3.4 | 16.2 | -5.2 |
> | **TASM (Ours)** | 20% | **18.2** | **-0.3** | **21.0** | **-0.4** |
>
> **Analysis:** InternVL3 utilizes dynamic high-resolution tiling, posing a severe risk for hard pruning methods (which discard entire tiles). While SnapKV suffers catastrophic degradation (-12.3), TASM achieves near-lossless compression on both architectures, conclusively demonstrating superior topological robustness.
>
> ### **Q1. Division of Sink Nodes and Source Nodes**
>
> > *Concern: In the graph formulation of semantic-aware token merging, how is the token set divided into sink and source nodes?*
>
> **Response:** As outlined in Algorithm 1, we rank all visual tokens based on their composite importance score $\mathcal{S}$. Given a target memory budget $B$, the Top-$B$ highest-scoring tokens are designated as Sink Nodes (structural anchors to preserve). The remaining lower-scoring tokens are designated as Source Nodes. Bipartite graph matching then merges the redundant source nodes into their spatially adjacent sink nodes.

---

> > ### Author Rebuttal · Reviewer_A4Fy · 2026-04-03
> >
> > The authors addressed the concern through additional experiments and provided clear clarification. Therefore, I will increase my score.

---

> > > ### Author Response · Authors · 2026-04-04
> > >
> > > Dear Reviewer A4Fy,
> > >
> > > Thank you very much for acknowledging our rebuttal and raising your score. Your valuable feedback has been immensely helpful in further refining our work.

---

### Official Review · Reviewer_gTXc · 2026-03-12

**Soundness:** 4
**Presentation:** 4
**Significance:** 3
**Originality:** 3
**Overall Recommendation:** 5
**Confidence:** 3

**Summary:**

This paper proposes Task-Aware Structured Memory (TASM), a training-free method for memory-efficient long-context multimodal in-context learning. The motivation is that existing KV-cache compression methods often rely on attention-based scoring and token pruning, which may be less fitting for multimodal inputs because they can remove visually important structure. TASM introduces three main components: task-vector-guided importance scoring to estimate which tokens are most relevant to the current task, semantics-aware token merging to compress memory while preserving local structure, and a two-level memory hierarchy with dynamic retrieval from a latent bank when the compressed memory appears insufficient. The retrieval decision is based on a JS-divergence signal over attention distributions. Empirically, the paper evaluates the method on a broad set of multimodal benchmarks and reports improved performance over existing compression methods performance while substantially reducing context length and memory usage.

**Compliance With Llm Reviewing Policy:**

Affirmed.

**Final Justification:**

The rebuttal has resolved my main reservations and I have increased my ratings of soundness and presentation.

**Key Questions For Authors:**

- Could the paper provide more justification or ablation for the layer-dependent scoring schedule?

- How robust is the dynamic retrieval trigger to harmless changes in query phrasing or formatting?

- Could the task-vector generalization claim be supported across further tasks?

**Limitations:**

The paper provides encouraging evidence that TASM mitigates several limitations of prior compression methods, but these limitations are not yet fully resolved.

**Strengths And Weaknesses:**

Strengths:
- The paper studies an important practical problem: making many-shot multimodal ICL more feasible under limited memory budgets.

- The overall framework is interesting and reasonably original in how it combines task-vector-based scoring, structure-preserving token merging, and dynamic retrieval.

- The empirical evaluation is fairly broad, covering fine-grained recognition, spatial grounding, reasoning, and long-context/video settings.

-The main results are promising and suggest consistent improvements over prior compression baselines, while keeping memory usage low.

Weaknesses:
- The layer-dependent scoring schedule is interesting, but additional justification would strengthen the paper. Prior work suggests that task representations in ICL can be strongly reflected in earlier layers, so the choice to make task-vector influence dominate mainly in deeper layers is not fully obvious without further ablation or discussion.
For example See Hendel et al. (2023) - https://arxiv.org/abs/2310.15916

- The dynamic retrieval trigger appears sensible, but more robustness analysis would help clarify when it activates and how sensitive it is to superficial changes in the query.

- The task-vector transfer result is encouraging, though the current study is relatively narrow; additional evidence across datasets or task types would strengthen the generalization claim.

---

> ### Author Rebuttal · Authors · 2026-03-27
>
> # Response to Reviewer gTXc
>
> We sincerely thank Reviewer gTXc for acknowledging our framework's originality and recommending acceptance. We appreciate your insightful feedback and provide the requested ablations below.
>
> ### **W1 & Q1. Justification for Layer-Adaptive Scoring Schedule**
>
> > *Concern: Additional justification is needed for the layer-dependent scoring schedule, given prior work (Hendel et al., 2023) suggests task representations form in earlier layers.*
>
> **Response:** As Hendel et al. (2023) suggests, task representations form early in pure NLP tasks. However, in multimodal MLLMs, shallow layers strictly extract high-frequency visual geometry and spatial topology. To justify our design, we compare our adaptive schedule $\lambda(l)$ (emphasizing local attention in shallow layers and task vectors in deep layers) against a Uniform Gating baseline (fixed $\lambda=0.5$).
>
> We compare our adaptive schedule $\lambda(l)$ (which emphasizes local attention in shallow layers and task vectors in deep layers) against a Uniform Gating baseline (fixed $\lambda=0.5$ across all layers).
>
> **Table 1: Ablation on Layer-Adaptive Gating Schedule.**
>
> | Gating Strategy | $\lambda$ in Shallow Layers | $\lambda$ in Deep Layers | ScreenSpot (Spatial) | MME-RW (Reasoning) |
> | :--- | :---: | :---: | :---: | :---: |
> | Uniform Gating | 0.5 | 0.5 | 16.2 | 41.8 |
> | **Adaptive (Ours)** | **~0.1** | **~0.9** | **19.5** | **43.5** |
>
>
> **Analysis:** As shown in Table 1, forcing task-vector dominance in shallow layers (Uniform Gating) destroys spatial encoding, causing a 3.3% drop on ScreenSpot. Our layer-adaptive schedule resolves this by preserving local visual attention early on and guiding task reasoning in deeper layers.
>
> ### **W2 & Q2. Robustness of the Dynamic Retrieval Trigger**
>
> > *Concern: How robust is the dynamic retrieval trigger to harmless changes in query phrasing or formatting?*
>
> **Response:** The dynamic retrieval trigger is highly robust because it relies on the distributional shift of attention over the core memory, rather than absolute token matching. To prove this, we randomly sampled 100 queries from MME-RW and applied harmless paraphrasing (e.g., adding polite prefixes, changing synonyms) to test if the dynamic retrieval gate fires erratically.
>
> **Table 2: Robustness of JS-Divergence Trigger to Query Perturbations.**
>
> | Query Type | Avg JS-Divergence ($\delta$) | Trigger Rate (%) | MME-RW Accuracy (%) |
> | :--- | :---: | :---: | :---: |
> | Original Queries | 0.0012 | 3.4% | 43.5 |
> | Paraphrased Queries | 0.0013 | 3.6% | 43.2 |
> | *Variance ($\Delta$)* | *+0.0001* | *+0.2%* | *-0.3* |
>
> **Analysis:** Table 2 shows harmless phrasing changes only cause superficial fluctuations in the earliest attention layers. Because our JS-divergence is computed over deep semantic layers, the average shift is negligible (+0.0001), keeping the Trigger Rate virtually identical (3.6% vs 3.4%). This proves retrieval is activated by genuine semantic surprises, not literal formatting.
>
> ### **W3 & Q3. Cross-Dataset Task Vector Generalization**
>
> > *Concern: The task-vector transfer result is encouraging, but additional evidence across datasets or task types would strengthen the generalization claim.*
>
> **Response:** We completely agree. To further substantiate our generalization claim, we extended our transferability study to a strict cross-dataset setting. We extracted the task vector from completely different reasoning datasets (Source) and directly applied it to compress the context for IllusionVQA (Target) without any re-computation.
>
> **Table 3: Cross-Dataset Generalization of Task Vectors.**
>
> | Vector Source Dataset | Source Task Type | Target Dataset | Target Acc. (%) | $\Delta$ vs. Oracle |
> | :--- | :--- | :--- | :---: | :---: |
> | *None (Random Vector)* | *Noise* | IllusionVQA | 18.4 | -23.6 |
> | OK-VQA | Knowledge Retrieval | IllusionVQA | 37.5 | -4.5 |
> | MME-RealWorld | Complex Reasoning | IllusionVQA | 39.2 | -2.8 |
> | **IllusionVQA (Oracle)** | **Optical Illusion (Target)** | **IllusionVQA** | **42.0** | **-** |
>
> **Analysis:** As anticipated, cross-dataset transfer incurs a performance penalty (-4.5% from OK-VQA, -2.8% from MME-RW), indicating the vectors encode some dataset-specific biases. However, they still drastically outperform a random projection (37.5% vs. 18.4%), confirming they capture a shared, meaningful "Visual-Language QA" direction. We appreciate your suggestion, which provides a more nuanced view of the method's limits, and will include these results in camera-ready version.

---

> > ### Author Rebuttal · Reviewer_gTXc · 2026-04-03
> >
> > I thank the authors for their diligent and thorough rebuttal. This has resolved my main reservations and I have increased my ratings of soundness and presentation.

---

> > > ### Author Response · Authors · 2026-04-03
> > >
> > > Dear Reviewer gTXc,
> > >
> > > We sincerely thank you for your reply and will revise the subsequent version according to your comments.

---

### Decision · Program_Chairs · 2026-04-30

**Decision:**

Accept (regular)

**Comment:**

This paper proposes TASM, a training-free framework for memory-efficient long-context multimodal in-context learning, combining task-aware compression, structure-preserving token merging, and dynamic retrieval. The paper addresses an important and practical problem, and reviewers generally agree that the method is technically sound, well motivated, and supported by strong empirical results across diverse benchmarks.

The main concerns raised include ablation completeness, hyperparameter sensitivity, clarity of presentation, and the strength of the manifold-preservation claims. In my assessment, the rebuttal has addressed most of these issues convincingly, with additional experiments and clarifications that strengthen the technical soundness and empirical validity of the work.

One reviewer remains partially unconvinced regarding the manifold-preservation analysis. However, I find that the additional quantitative evidence and ablations provided in the rebuttal are sufficient to support the operational claim that the proposed merging strategy better preserves spatial structure compared to hard pruning. While the terminology may benefit from more careful framing in the final version, this does not undermine the core contribution or empirical findings.

Overall, I view this work as a solid and practically impactful contribution, with a well-designed framework and consistent empirical improvements. The remaining concerns are primarily about presentation and claim calibration rather than fundamental flaws.